# Interannual variabilities, long-term trends, and regulating factors of low-oxygen conditions in the coastal waters off Hong Kong

Zheng Chen[1], Bin Wang[4], Chuang Xu[5], Zhongren Zhang[1,6], Shiyu Li[1], Jiatang Hu[1,2,3,*]

[1]School of Environmental Science and Engineering, Sun Yat-sen University, Guangzhou, 510275, China
[2]Guangdong Provincial Key Laboratory of Environmental Pollution Control and Remediation Technology, Guangzhou, 510275, China
[3]Southern Marine Science and Engineering Guangdong Laboratory (Zhuhai), Zhuhai, 519000, China
[4]Department of Oceanography, Dalhousie University, Halifax, Nova Scotia, B3H 4R2, Canada
[5]Center for Water Resources and Environment, Sun Yat-sen University, Guangzhou, 510275, China
[6]Guangdong Zhihuan Innovative Environmental Technology Co., Ltd, Guangzhou, 510030, China

*Correspondence to:* Jiatang Hu (hujtang@mail.sysu.edu.cn)

**Abstract.** The summertime low-oxygen conditions in the Pearl River Estuary (PRE) have experienced a significant expansion in spatial extent associated with notable deoxygenation in recent decades. Nevertheless, there is still a lack of quantitative understanding of the long-term trends and interannual variabilities in oxygen conditions in the PRE as well as the driving factors. Therefore, the long-term deoxygenation in a subregion of the PRE (the coastal waters off Hong Kong) was comprehensively investigated in this study using monthly observations during 1994-2018. To evaluate the changes in scope and intensity of oxygen conditions, an indicator (defined as the Low-oxygen Index, LOI) that integrates several metrics related to low-oxygen conditions was introduced as the result of a principal component analysis (PCA). Moreover, primary physical and biogeochemical factors controlling the interannual variabilities and long-term trends in oxygen conditions were discerned, and their relative contributions were quantified by the multiple regression analysis. Results showed that the regression models explained over 60% of the interannual variations in LOI. Both the wind speeds and concentrations of dissolved inorganic nitrogen (DIN) played a significant role in determining the interannual variations (by 39% and 49%, respectively) and long-term trends (by 39% and 56%, respectively) in LOI. Due to the increasing nutrient loads and alterations in physical conditions (e.g. the long-term decreasing trend in wind speeds), coastal eutrophication was exaggerated and massive marine-sourced organic matter was subsequently produced, thereby resulting in an expansion of intensified low-oxygen conditions. The deteriorating eutrophication has also driven a shift in the dominant source of organic matter from terrestrial inputs to in situ primary production, which has probably led to an earlier onset of hypoxia in summer. In summary, the Hong Kong waters have undergone considerable deterioration of low-oxygen conditions driven by substantial changes in anthropogenic eutrophication and external physical factors.

**Keywords:** Dissolved oxygen; Low-oxygen conditions; Interannual variations; Long-term deoxygenation; Pearl River Estuary

## 1. Introduction

Dissolved oxygen (DO) plays a vital role in maintaining the good functioning of aquatic ecosystems. Hypoxia (DO < 2 mg/L) could lead to a marked reduction in habitat for aquatic organisms (Ludsin et al., 2009) and imposes detrimental effects on ecosystem community structure and energy flow (Diaz and Rosenberg, 2008). In recent decades, long-term exacerbation on hypoxia in terms of its spatial extent and intensity has been documented in estuaries and coastal waters worldwide, including the Baltic Sea (Conley et al., 2011; Meier et al., 2019), the northern Gulf of Mexico (Obenour et al., 2013; Laurent and Fennel, 2019), Chesapeake Bay (Li et al., 2016; Ni et al., 2020), the Yangtze River Estuary (Zhu et al., 2011; Zhang et al., 2021), and the Pearl River Estuary (Li et al., 2020; Hu et al., 2021). In addition, changes in the phenology of hypoxia were also reported. For example, in Chesapeake Bay, hypoxic volume has shown a significant increase in early summer but a slight decrease in late summer since 1985 (Murphy et al., 2011; Testa et al., 2018) . Zhou (2014) also found that the timing of maximum hypoxic volume in Chesapeake Bay was advanced from late July to early July during 1985-2010.

A great number of studies have indicated that the exacerbation of hypoxia in coastal systems was closely related to human activities, such as urbanization and industrialization (Breitburg et al., 2018). Due to the anthropogenic influence, massive organic matter and nutrients were discharged into estuaries and coastal waters. Terrestrial organic matter could lead to intense microbial respiration (Rabalais et al., 2010) and excessive nutrient inputs could further stimulate the growth of phytoplankton and exacerbate eutrophication, with a dramatic increase in oxygen demand from marine-sourced organic matter (Fennel and Testa, 2018). Meanwhile, physical processes such as stratification (Rabalais et al., 1991), convergence and migration of water masses (Li et al., 2021), and upwelling (Feng et al., 2014) could regulate the spatial extent and intensity of hypoxia as well. These processes are closely linked to wind forcing and freshwater discharge (Feng et al., 2012; Yu et al., 2015). In general, the physical and biogeochemical processes exert joint impacts on the generation and development of hypoxia, but different mechanisms may predominate in different systems due to their distinctive natural conditions (e.g. topography) and pressure from anthropogenic pollution. Ni (2020) quantified the contributions of estuary warming, sea level rise, and nutrient load reduction to the long-term changes in hypoxia in Chesapeake Bay through numerical simulation experiments, suggesting that warming was the dominant factor. Forrest (2011) investigated the effects of various processes on the interannual variations of hypoxia in the northern Gulf of Mexico by statistical methods and pointed out that the east-west winds and nutrient loads each accounted for a considerable contribution. While in the Yangtze River Estuary, studies showed that vertical density stratification, which was heavily influenced by a combination of freshwater inputs, various water masses, and winds, was the key factor controlling the interannual changes in hypoxia (Chi et al., 2020).

With the rapid socioeconomic development, the Pearl River Estuary (PRE) has received a large amount of pollutants and nutrients, resulting in a series of environmental problems, including eutrophication, red tide, and hypoxia (Dai et al., 2008; Li et al., 2020). Since the 1980s, low-oxygen (DO < 4 mg/L) and hypoxic conditions have been reported in the upper reach of Lingdingyang Bay (Li et al., 2020; Cui et al., 2018; Hu et al., 2021), Modaomen Bay, Huangmaohai Bay (Su et al., 2017; Shi et al., 2019; Wang et al., 2017; Zhang and Li, 2010) and coastal waters adjacent to Hong Kong (Yin et al., 2004; Su et al.,

2017; Shi et al., 2019). Previous studies have shown that hypoxia in the PRE typically occurred in the bottom waters during summer (Yin et al., 2004), driven by strong stratification and sediment oxygen consumption (Zhang and Li, 2010; Wang et al., 2017). Due to relatively shallow topography, short water residence time (Rabouille et al., 2008) and short maintenance of stratification (Luo et al., 2009; Lu et al., 2018) hypoxia in the PRE appeared to be episodic and localized (Rabouille et al., 2008). However, this long-standing point of view has been challenged by recent observations showing the emergence of large low-oxygen and hypoxic extents. The area affected by low oxygen in the bottom waters of the PRE was estimated to be around 1,000 km$^2$ in 2010 (Wen et al., 2020) and ~1,500 km$^2$ in 2015 (Li et al., 2018). With the increasing availability of observations, an apparent expansion of hypoxia with large interannual variations has been revealed from the data during 1976-2017 (Hu et al., 2021). Nevertheless, due to the scarcity of observations in both time and space and significant differences in sampling periods and locations (sometimes the water quality measurement methods as well) between available datasets, a clear understanding of the long-term trend and interannual changes in hypoxia in the PRE as well as the associated drivers is still lacking, especially from a quantitative perspective.

In this study, we utilize observational oxygen and related data collected by the Hong Kong Environmental Protection Department (HKEPD) at certain coastal sites off Hong Kong (see details in section 2.1 below), which have significant merits in terms of temporal coverage (~30 years) and consistency of sampling locations, to perform a quantitative analysis on the long-term oxygen changes (trend and interannual variability) in the region. Moreover, we also aim to discern the key factors controlling the interannual variability and long-term trends in the low-oxygen conditions and to quantify the relative contribution of each primary factor using multiple regression models (Murphy et al., 2011; Forrest et al., 2011; Wang et al., 2021). It is important to note that the HKEPD data with good spatiotemporal continuity allowed us to better estimate the long-term deoxygenation in the coastal waters off Hong Kong, which was close to a hotspot area of low-oxygen conditions in the eastern PRE (Hu et al., 2021) and subject to frequent occurrences of low-oxygen and hypoxic events as well (Yin et al., 2004; Su et al., 2017; Shi et al., 2019). In addition, previous studies have showed that the dominant deoxygenation mechanisms varied between subregions in the PRE; for instance, the low-oxygen conditions in Modaomen Bay were primarily determined by terrestrial pollutant inputs (Li et al., 2020; Wang et al., 2017; Wang et al., 2018), whereas those in the coastal waters off Hong Kong were largely controlled by the joint effect of physical processes (e.g. convergence of water masses (Li et al., 2021)) and eutrophication (Qian et al., 2018). Therefore, the extensive investigation on deoxygenation performed here for the Hong Kong waters is a significant supplement to the understanding of low-oxygen conditions for the whole PRE.

## 2. Materials and methods

### 2.1 Data sources

Monthly monitoring data from the HKEPD at 10 stations (Figure 1) in the coastal waters off Hong Kong (113.8~114.5° E, 22.1~22.6° N) was chosen for formal analysis. Specifically, the data in use include vertical profiles of DO, temperature, salinity, dissolved inorganic nitrogen (DIN), and chlorophyll *a* (Chl *a*) concentrations measured in the water columns during

1994-2018 as well as total organic carbon (TOC) and total nitrogen (TN) measured in the sediments during 1998-2018. The survey stations can be divided into three subregions: (1) the northwestern subregion, including stations NM5 (with water depth of 20 m), NM6 (5 m), and NM8 (8 m); (2) the southern subregion, including stations SM20 (7 m), SM17 (12 m), SM18 (21 m), and SM19 (24m); and (3) the eastern subregion, including stations MM8 (31 m), MM13 (28 m), and MM14 (25 m). Water samples were collected from the surface (1 m below the sea surface), middle (half of the depth at each station), and bottom (1 m above the sediments) layers, respectively. Details on the sampling procedures and measurements were described in Xu et al (2010).

In addition, the monthly data of wind speeds and directions used for analysis were estimated using the daily wind observations during 1994-2018 provided by the Waglan Island automatic weather station (Figure 1) of the Hong Kong Observatory. It should be noted that the duration of southwestern winds was defined as the number of its occurrence in days during summer. As for the freshwater inputs from the Pearl River, the monthly data during 1994-2018 were calculated using the discharge data obtained from three major hydrological stations (i.e. Gaoyao, Shijiao, and Boluo) of the Pearl River Water Resources Commission of the Ministry of Water Resources.

**2.2 Statistical methods**

Several metrics, including the cross-sectional area and the layer thickness of low oxygen (DO < 4 mg/L), oxygen deficiency (DO < 3 mg/L) and hypoxia (DO < 2 mg/L) as well as the mean and minimum DO concentrations in the bottom waters, were used to depict the oxygen conditions in the region. Firstly, the observed DO profiles were interpolated by "Natural-Neighbor" method through MATLAB along the three subregions with a grid resolution of 600 m (distance) × 0.3 m (depth). The total areas of DO below 4 mg/L, 3 mg/L, and 2 mg/L were then calculated as the cross-sectional areas of low oxygen, oxygen deficiency, and hypoxia, respectively. The associated layer thickness was defined as the averaged thickness of the grids with DO below the corresponding levels (i.e. 4 mg/L, 3 mg/L, and 2 mg/L). Regarding the island between stations NM8 and SM20, the spatial interpolations were performed directly with all the observed data and then the areas covered by the island were masked out roughly based on its size (Figures 2, 3, A1), as the topographic data of the island was not available. Such a treatment has little influence on the estimation of vertical low-oxygen areas because low-oxygen conditions were seldom found in stations NM8 and SM20. Moreover, the same treatment procedure was applied to the data in each month of 25 years to generate an interpolation set for every month, making it consistent when investigating the interannual variations in low-oxygen conditions.

In order to investigate the main variation (interannual changes) of oxygen conditions, we have introduced an indicator integrating the above metrics (except the hypoxic area and thickness, Table A1) through the PCA (principal component analysis) technique, which can reduce the dimensionality of a dataset to make it more interpretable with minimum information loss (Cadima et al., 2016). The two metrics related to hypoxia were excluded from PCA because the occurrence of hypoxia was relatively rare and its interannual variation was not as significant as that of low oxygen and oxygen deficiency. The results of PCA analysis (Table A2) showed that the first component explained most of variance (86.40%) for the six input variables,

while the remaining components explained less variance (13.60%). The first component was highly correlated with the

130 interannual variations of the cross-sectional areas (with a correlation coefficient r of 0.96, p < 0.01) and the thickness (r = 0.96, p < 0.01) of low oxygen as well as the bottom DO concentrations (r = -0.90, p < 0.01), and it was thereafter referred as Low-oxygen Index (LOI, Equ. 1) to describe the interannual severity of low-oxygen conditions comprehensively.

$$LOI = -0.40 \times DO_{mean} - 0.39 \times DO_{min} + 0.42 \times Area_4 + 0.41 \times Area_3 + 0.42 \times Thickness_4 + 0.41 \times Thickness_3 \quad (1)$$

where $DO_{mean}$ and $DO_{min}$ represent the mean and the minimum DO concentrations in the bottom waters, respectively; $Area_4$

($Area_3$) and $Thickness_4$ ($Thickness_3$) represent the cross-sectional area and the thickness of low oxygen (oxygen deficiency), respectively.

As the low-oxygen conditions within Hong Kong waters were jointly affected by physical and biogeochemical processes, we attempted to quantify the relative contributions of multiple relevant factors including wind, freshwater, water temperature, and nutrients to interannual variability and long-term trends of the oxygen conditions through multiple regression. As for the

140 selection of the wind variable in use, the daily wind data were processed into monthly average wind speed (WS), southwestern wind duration (SWWD), southwestern wind cumulative stress (SWCS), and southeastern wind cumulative stress (SECS) in summer (June-August) to examine the effect of wind speed and direction (Figure A2). Then, a suite of multiple regressions was carried out to fit the LOI for each wind-related variable. As shown in Table A3, the fitting effect of LOI was better when using WS, which also has the highest correlation with LOI among the wind-related variables, revealing that WS explained the

145 most interannual variation of LOI among the wind-related factors. Therefore, WS was eventually adopted to be the wind-related input variable in the multiple regression with freshwater discharge (flow), the monthly spatial-average of bottom temperature (T), and surface DIN in the summer. The resulting regression coefficients were then standardized by multiplying the ratio between the standard deviation of each input variable (e.g. WS) and the standard deviation of LOI to evaluate their interannual contributions (Equ. 2).

$$Cst_i = C_i \times \frac{SD_i}{SD_{LOI}} \quad (2)$$

Where $Cst_i$ and $C_i$ represent the regression coefficients of WS, flow, T, and DIN after and before standardization, respectively; $SD_i$ represent the standard deviation of WS, flow, T, and DIN; $SD_{LOI}$ represents the standard deviation of LOI.

In addition, the dataset was randomly split into a training dataset (70%) and a testing dataset (30%) in order to provide a more robust data fitting with estimates on the uncertainties arising from different data selections. Consequently, over 480,700

combinations of training and testing datasets were generated randomly from this splitting process and were used to build up a variety of regression models. Coefficient of determination ($R^2$) was used to measure the fitting effect in training and testing datasets. Of all the established models, the fitting effect of training datasets (e.g. $R^2_{train}$) and coefficients of the four variables were similar, but the predictive skills in testing dataset (e.g. $R^2_{test}$) varied in a large range (Figure A3, Table A4). Besides, larger standard deviation occurred in coefficients in cases with worse testing effects. To provide a more robust estimation for

the fitting, only those with $R^2$ over or equal to 0.6 both for the training and testing datasets were selected to quantify the impact of each input variable according to their regression coefficients on average (Figure A4). Furthermore, based on the selected

models (with $R^2 \geq 0.6$ for both datasets), we also set up four sensitive experiments in which the long-term trend of each input variable was removed and only interannual fluctuations were retained. The LOI was then re-calculated in each scenario and its change relative to the original LOI was used to assess the impact of each variable to the long-term oxygen trend.

## 3. Result

### 3.1 Seasonal and interannual variabilities in water quality variables in the coastal waters off Hong Kong

### 3.1.1 Hydrological and eutrophication parameters

Significant seasonal variations could be found for the hydrologic settings (Figure 2). In winter (December-February), temperature generally exhibited low levels, with climatological mean values of 18.62 °C and 18.54 °C during 1994-2018 in the surface (Figure 2a) and the bottom waters (Figure 2b), respectively; salinity reached high values due to the invasion of shelf saline waters, with means of 32.05 PSU at the surface (Figure 2c) and 32.44 PSU at the bottom (Figure 2d). Small differences of temperature and salinity between the surface and the bottom layers in winter indicated that the water column was well mixed (with mean vertical density differences of 0.33 kg/m$^3$; Figure 2e). By comparison, temperature and salinity in summer (June-August) showed larger vertical gradients and interannual variability. The summertime temperature fluctuated between 28.21±1.19 °C (i.e. climatological mean±one standard deviation) at the surface, which was markedly higher than that at the bottom (24.93±2.14 °C). As affected by massive freshwater inputs from the Pearl River, salinity in summer was much lower than that in winter and displayed pronounced vertical differences with 22.86±7.53 PSU at the surface and 30.90±5.26 PSU at the bottom, respectively. Consequently, strong water stratification prevailed in summer, where the vertical density differences (Δρ) fluctuated between 7.26±4.54 kg/m$^3$ (Figure 2e).

DIN and Chl *a* are two important parameters related to eutrophication and they both showed remarkable changes over time (Figure 3a-d). In winter, the concentrations of DIN and Chl *a* were generally low, with climatological means of 0.19 mg/L (surface) and 0.16 mg/L (bottom) for DIN and means of 2.45 μg/L (surface) and 2.04 μg/L (bottom) for Chl *a*. While in summer, DIN and Chl *a* reached comparatively high levels with significant interannual variability. Overall, the DIN concentrations fluctuated between 0.56±0.50 mg/L at the surface (Figure 3a), which was higher that at the bottom (0.28±0.34 mg/L; Figure 3b). Chl *a* also showed considerable vertical differences with 8.56±9.30 μg/L at the surface (Figure 3c) and 2.46±4.13 μg/L at the bottom (Figure 3d).

In terms of spatial distributions, distinct differences were observed for the hydrological and eutrophication parameters among the three subregions investigated. Due to the profound influence of river discharges, temperature/salinity in the northwestern subregion (NM5-NM8, closer to the river outlets) was noticeably higher/lower when compared to the other two (Figure 2a-d), varying by 28.61±1.09 °C/14.63±6.23 PSU at the surface in summer. Meanwhile, the DIN concentration in the northwestern subregion was the highest (Figure 3a-b), reaching up to 1.17±0.40 mg/L at the surface. On the contrary, the eastern subregion (MM8-MM14), which was farthest to the river outlets and more heavily affected by the shelf water, had the

lowest temperature (27.85±1.27 °C), highest salinity (29.22±3.10 PSU) and lowest DIN concentration (0.14±0.13 mg/L) in the surface waters. As for Chl *a* (Figure 3c-d), the highest level appeared in the southern subregion (SM17-SM20, with 10.19±8.86 μg/L at the surface in summer), while the lowest one was found at the northwestern subregion (with 6.82±10.67 μg/L at the surface).

### 3.1.2 Dissolved oxygen and low-oxygen conditions

DO concentrations exhibited significant seasonal and interannual variations in both layers (Figure 3e-f). The DO concentrations maintained at higher levels during winter (with 7.07±0.99 mg/L and 7.27±0.87 mg/L in the surface and the bottom waters over 1994-2018, respectively) and dropped to a level of 6.91±1.71 mg/L at the surface and 4.42±1.37 mg/L at the bottom in summer. Statistic results showed that low-oxygen events mainly appeared in the bottom waters of summer, which had much higher occurrences of DO < 4 mg/L and DO < 2 mg/L compared to other seasons and other layers (Figure A5). In addition, the summertime DO minimum at the bottom (Figure 4a) fluctuated between 2.28±0.89 mg/L, further indicating the water quality deterioration with severe oxygen deficits in the Hong Kong waters. Among the three subregions, the northwestern and the southern ones had relatively lower bottom DO levels (with 4.56±1.56 mg/L and 4.14±1.45 mg/L, respectively) and considerably higher occurrences of low-oxygen conditions (with 38.76% and 49.32%, respectively) compared to the eastern subregion (with DO of 4.68±0.93 mg/L and occurrence of 17.24%; Figure A5).

In addition to the DO levels, we also investigated the interannual changes in the summertime low-oxygen conditions in terms of areal extents (vertical profiles), thickness, and the LOI as defined in section 2.2 (Figure 4). Significant interannual fluctuations were found for all these metrics; for example, the area and thickness affected by low oxygen fluctuated between $(3.35\pm2.38)\times10^5$ m$^2$ and 5.66±4.01 m, respectively, while those for oxygen deficiency were $(7.13\pm8.37)\times10^4$ m$^2$ and 1.20± 1.41 m. Low-oxygen and hypoxic conditions were more severe in the years such as 2007, 2011, and 2017, as indicated by the high LOI values. In particular, the year 2011 had the largest low-oxygen area ($\sim7.66\times10^5$ m$^2$) and the lowest DO concentration ($\sim0.40$ mg/L) over the past 25 years, thus possessing the highest LOI; it could be observed that the low-oxygen waters almost occupied the entire middle-to-bottom layers across all the sites during this period (Figure A1). On the other hand, hypoxic conditions were absent in some years (e.g. 2004, 2006, and 2018), where the water column resided in a comparatively well-oxygenated status (Figure A1); the corresponding LOI in these hypoxia-relief years fell to large negative values (Figure 4d).

### 3.2 Long-term trends of low-oxygen conditions in the coastal waters off Hong Kong

Despite the large DO fluctuations by years, a clear deoxygenation trend could be observed in summer over the past 25 years, showing a long-term decline in the DO concentrations associated with increases in the areas and occurrences affected by low oxygen (Figure 4). More specifically, before 2000 the spatially-averaged DO concentrations in the bottom waters exceeded 4 mg/L and low-oxygen conditions were seldom observed (Figure 4a), while the DO minimums were all above 2 mg/L (i.e. no hypoxic events occurred). However, since 2000 the occurrences of low oxygen and hypoxia have become more

frequent, with a significant growth in the LOI and its related metrics, confirming the exacerbation of low-oxygen conditions in the Hong Kong waters.

To further quantify the intensity of long-term deoxygenation in summer, linear regressions were performed for the DO concentrations in different layers and in different subregions and also for the areal extents of low-oxygen conditions during 1994-2018 (Figure 5). As shown, apparent declining trends were found for the DO series both at the surface (although not significant, Figure 5a) and the bottom (Figure 5b). For the bottom waters, the averaged DO concentrations displayed a decreasing pattern with a rate of 0.03 mg/L per year (equivalent to approximately 0.7% of the climatological DO mean at the bottom), while the DO minimums showed a more significant decline with a rate of 0.08 mg/L per year (~3.5% of the climatological mean of the bottom DO minimums). It was also noted that the intensity of deoxygenation varied between subregions (Figure 5c-h). As for the bottom DO concentrations, the most significant decrease was found in the eastern subregion (with a deoxygenation rate of 0.05 mg/L per year, Figure 5h), while the most significant decline in the DO minimum appeared in the southern subregion (with a rate of 0.08 mg/L per year, Figure 5f). Likewise, significant increasing trends were also found for the areas of low oxygen and oxygen deficiency (Figure 5i-j), showing an annual growth rate at $1.95 \times 10^4 \, \mathrm{m}^2$ and $4.75 \times 10^3 \, \mathrm{m}^2$, respectively. Regarding the changes in LOI, it had a growth rate of 0.20 per year, which corresponds to an increasing rate of $1.99 \times 10^4 \, \mathrm{m}^2$ in the low-oxygen area and a declining rate of 0.07 mg/L in the DO minimum.

Furthermore, the long-term oxygen changes varied between different months of the summer season as well (Figure 6). It could be seen that the decreasing magnitude of the averaged DO concentration was close to each other for all the summer months, while the decline in the DO minimum was most pronounced in July (with a decreasing rate of 0.10 mg/L per year, Figure 6c), followed by that in August (0.06 mg/L per year, Figure 6e). In fact, the long-term changes in the DO minimum had different patterns in July and August. As for July, the DO minimum generally showed a consecutive decrease over the past 25 years (Figure 6d). While in August, the DO minimum experienced a rapid decline with a rate of 0.14 mg/L per year during 1994-2011, which was higher than that in July during the same period (0.11 mg/L per year), but subsequently undertook a recovery from the hypoxic conditions since 2012 (Figure 6f). Along with such distinctive intra-seasonal patterns, an interesting phenomenon was also noticed: hypoxic events were present mostly in August prior to 2012 (e.g. in 2007 and 2010-2011; no hypoxia was found in July during the same period) but only in July instead since 2012 (e.g. in 2014 and 2016-2017), as shown in Figure 6. This finding implied a potential shift in the onset of hypoxia generation from August to July, i.e. an earlier timing for the arrival of the summertime hypoxia. Accordingly, distinct changes were found for the areas affected by hypoxia in the two periods around 2012. The hypoxic area estimated in July increased from zero during 1994-2011 to $(5.42 \pm 8.77) \times 10^3 \, \mathrm{m}^2$ during 2012-2018, whereas the hypoxic area in August decreased from $(0.89 \pm 2.82) \times 10^4 \, \mathrm{m}^2$ to zero.

## 4. Discussion

### 4.1 Primary factors controlling the interannual variabilities in low-oxygen conditions

As shown above, significant interannual variabilities were observed in the spatial extent (e.g. cross-sectional area) and intensity of oxygen conditions (e.g. the mean bottom DO concentrations). Such variabilities were largely influenced by multiple physical and biogeochemical factors, including wind forcing, freshwater discharge, water temperature and nutrient loads. These processes jointly act to affect density stratification (Yu et al., 2015), water residence time (Li et al., 2021) and temporal and spatial distributions of eutrophication parameters (Cui et al., 2018). As described in section 2.2, four important

influential factors (i.e. WS, flow, T, and DIN) were used to predict the interannual variations in LOI by the multiple regression models, in which there have been 56,010 cases (~12% of the total, Figure 7) with $R^2 \geq 0.6$ both in the training dataset (mean $R^2$ of 0.64) and the testing dataset (mean $R^2$ of 0.70). The standardized coefficients (mean±standard deviation) for these well-performing regression cases were given as follows:

$$LOI = -(0.39 \pm 0.12) \times WS - (0.14 \pm 0.12) \times flow - (0.11 \pm 0.08) \times T + (0.49 \pm 0.12) \times DIN \qquad (3)$$

As denoted by the regression coefficients, wind forcing has exerted a significant impact on the interannual changes in LOI, with a relative contribution of 39%±12% to the LOI variability explained. Its importance could also be evidenced by the significant negative correlation between WS and LOI (r = -0.67, p < 0.01, Figure 8), suggesting that calm winds were beneficial to low-oxygen conditions. In most cases (e.g. in Modaomen Bay in the PRE and the northern Gulf of Mexico), strong winds could break down stratification in the water column (Rabalais et al., 1991; Feng et al., 2012), which was conducive to water

mixing and atmospheric reoxygenation (Rabalais et al., 1991). However, the weak correlation between WS and Δρ (Figure 8) indicated that the wind forcing may control hypoxia through other alternative mechanisms. Actually, weak winds in combination with flow convergence induced by wind-driven circulation could contribute to long water residence time and nutrient accumulation in the eastern PRE and thus favored the phytoplankton blooms (Li et al., 2021). This could be supported by the significant negative correlation between WS and Chl *a* (r = -0.62, p < 0.01). In contrast, the wind direction showed less

significant effect on the interannual variability in low-oxygen conditions, as suggested by the comparatively poor performance in the LOI fitting and the weaker correlations of the wind direction-related variables with LOI (Table A3). It was noted that the monthly average wind direction in summer were generally southerly with small changes (mostly varying between 150° and 200°, Figure A2). Overall, our results indicated that the wind speed played a more important role in regulating the low-oxygen conditions in the coastal waters off Hong Kong from an interannual perspective, although the wind direction could

significantly influence the short-term generation and development of low-oxygen conditions by modulating the Pearl River plume and material fluxes (Yin et al., 2004; Li et al., 2021). With respect to the DIN concentrations, it played a vital role in determining the interannual variabilities of the oxygen conditions, with a contribution up to 49%±12%. It has been widely recognized that eutrophication stimulated by anthropogenic nutrient inputs could provide a large quantity of depositing detritus and subsequently led to substantial oxygen depletion and occurrence of low-oxygen events (Rabalais et al., 2010; Fennel and

Testa, 2018); for example, in the northern Gulf of Mexico (Feng et al., 2012; Forrest et al., 2011) and Chesapeake Bay (Wang

et al., 2015), the interannual hypoxic areas in summer were directly regulated by the nutrient levels. Similar situation was found in the PRE (Li et al., 2020) and Hong Kong waters, as confirmed by the significant positive correlation between DIN and LOI ($r = 0.65$, $p < 0.01$). Collectively, DIN and WS were identified as the two key factors controlling the interannual changes in low-oxygen conditions.

Compared to WS and DIN, the freshwater discharges (flow) had a much smaller contribution (~14%±12%) to the variations in LOI. Generally speaking, large freshwater inputs tend to enhance the intensity of water stratification and facilitate the generation of hypoxia (Rabalais et al., 1991). However, we found a negative correlation between flow and LOI ($r = -0.45$, $p < 0.05$, Figure 8), implying that the effect of freshwater discharges on low-oxygen conditions might involve more complex mechanisms and act through indirect pathways. Due to its long distance from the river outlets of the Pearl River, the coastal

waters off Hong Kong were relatively less influenced by terrestrial inputs (Yu et al., 2020) and the effect of freshwater discharge and its carrying organic matter in this area was not as significant as that in other subregions (e.g. the upper reach of Lingdingyang Bay and the western PRE). Nevertheless, freshwater discharge in combination with the wind-driven circulation could significantly affect the water residence time (Sun et al., 2014) and nutrients accumulation in the Hong Kong waters (Li et al., 2021). Specifically, the weakened discharge could prolong the retention of nutrients and thereby stimulate local

productions of organic matter in the region (Li et al., 2021), which ultimately promoted oxygen depletion. Regarding the water temperature, previous studies have showed that it could exert significant influence on coastal hypoxia largely by regulating water stratification intensity, oxygen solubility, and microbial respiration rate (Breitburg et al., 2018). However, our results showed that the contribution of water temperature (T) to the LOI changes (~11%±8%) was not significant in the Hong Kong waters, as revealed by its weak correlation with LOI as well. Given the fact that the Hong Kong waters are a region heavily

affected by human activities, the effect of temperature (e.g. global warming) might be more significant in the region with larger geography scale. Overall, the role of temperature and freshwater discharges in regulating the interannual oxygen variability in the Hong Kong waters appeared to be secondary.

## 4.2 Drivers of the long-term deoxygenation trend

The data over the past 25 years showed that the coastal waters off Hong Kong has experienced a notable long-term oxygen

decline, especially for the DO minimum in the bottom waters. Based on the observed deoxygenation rate, the bottom DO minimum was expected to decrease by approximately 15%-70% in 5-20 years (reaching a level of 0.4-1.6 mg/L) compared to the climatological mean of 1994-2018. The impacts of influential factors on the long-term deoxygenation trend were then evaluated using the regression models mentioned in section 4.1 and quantified by the relative changes of LOI in the sensitive experiments (see details in section 2.2) compared to the original one. It was noted that WS exhibited a decreasing trend of 0.03

m/s per year ($p < 0.05$, Figure 9a) within the coastal regions off Hong Kong over the past 25 years, while similar situation was also found in the Pearl River Basin (Zhang et al., 2019) and the northern South China Sea (Gao et al., 2020) due to the long-term climate changes (Xu et al., 2006; Zhang et al., 2009; Chen et al., 2020). Meanwhile, DIN showed an increasing trend with a rate of 0.01 mg/L per year ($p < 0.01$, Figure 9e). The growth in DIN and decline in WS have led to a 56%±10% and

39%±14% increase in LOI (Table 1), respectively, indicating that DIN and WS were the main driving factors for the long-term deoxygenation. On the other hand, significant long-term trends were also found for the freshwater discharges (with a decreasing rate of $4.19\times10^2$ m$^3$/s per year, Figure 9b) and water temperature (with an increasing rate of 0.06 °C per year, Figure 9d), but their impacts were relatively small, resulting in a 16%±14% increase and 11%±9% decrease in LOI, respectively.

Despite the different influences of the factors mentioned above, they were likely to impose synergetic impacts on the low-oxygen conditions by aggravating eutrophication as discussed earlier; it could be observed that the long-term growth in Chl $a$ (with a rate of 0.15 µg/L per year, Figure 9f) matched well with the increase in LOI. Specifically, the significant increase in phytoplankton biomass was primarily due to the combined effects of more stable water-column condition and longer residence time facilitated by the weaken wind forcing and river discharges, higher nutrient levels, and lower water turbidity (Figure 9g) in recent years. Consequently, the elevated organic matter through phytoplankton primary production would lead to strong oxygen consumption, thereby contributing to an expansion of low-oxygen conditions in terms of areal extent and intensity.

Moreover, with massive algal fragments provided by primary production, the composition of organic matter in the coastal waters off Hong Kong has probably changed and would cause substantial changes in the timing of hypoxia generation. As noted in section 3.2, the onset of hypoxia was observed to shift from August to July around 2012. To explore this issue, we first used the ratio of TOC to TN measured in the sediments to estimate the main source of organic matter, with values of 14-30 pointing to a terrestrial source and values of 4-10 indicating a marine source from in-situ production (Bordovskiy, 1965; Meyers and Ishiwatari, 1993). It is clear that the TOC:TN showed a significant decreasing trend and was mostly below 10 since 2012 (Figure 9h). This implied a shift in the dominant source of organic matter from terrestrial inputs to local production (marine-sourced). As such, oxygen consumption became faster because the marine-sourced organic matter was fresher and more active (Raymond and Bauer, 2001) and therefore the time required to reach hypoxia would be shortened. Furthermore, changes in the physical conditions provided sufficient time for more thorough decomposition of organic matter in July, which left less organic matter for August and thus weakened the deoxygenation therein.

Similarly, the long-term oxygen changes in terms of the areal extents and arrival timing of hypoxia have also been found in other coastal systems. For example, in Chesapeake Bay, sea level rise and elevated freshwater discharges would lead to an approximately 10%-30% increase in hypoxic volume between the late 20[th] and the mid-21[st] centuries (Ni et al., 2019), while the increase in water temperature would cause hypoxia to develop 5-10 days earlier in ~30 years (Ni et al., 2020). In the northern Gulf of Mexico, the growth in riverine nutrient inputs would result in an increase in the frequency of hypoxia occurrence by 37% (Justić et al., 2003). While in the Hong Kong waters, low-oxygen conditions would develop into hypoxic conditions in two decades with larger areal extent and earlier arrival ascribed to the ongoing alterations in physical conditions and nutrients as mentioned earlier. This inference was based on the assumption that the external factors (e.g. wind speed, DIN, discharges) would change at the same rates as those in the past 25 years. Although the real situation would be more complicated and compounded by factors such as the implementation of management and non-linear changes in climatic factors, our findings still served as an alarming signal that changes in wind and freshwater discharges could cancel out potential benefits of nutrient

management. To this end, it is of great importance to conduct long-term and more intensive control on nutrient inputs in order to mitigate the low-oxygen conditions in the region.

## 5. Conclusion

We have comprehensively investigated the spatiotemporal characteristics of DO and various related water quality variables in the coastal waters off Hong Kong and found that low-oxygen conditions occurred mostly in the bottom waters of summer, with significant interannual variability and an apparent deoxygenation trend over the past 25 years. We have also quantified the contribution of each primary factor by statistic methods and found that the increasing DIN levels and the decreasing wind speeds, both of which would eventually lead to the intensification of eutrophication, contributed most to the

interannual variations and long-term trend in LOI. Therefore, more marine-sourced organic matter was produced by the elevated primary production, leading to an exacerbation in low-oxygen conditions with larger areal extents as well as a potential earlier onset of the summertime hypoxia. By comparison, the freshwater inputs and water temperature had relatively small impacts on the long-term changes in LOI. To sum up, this study has shown that oxygen conditions in the coastal waters off Hong Kong have been deteriorating under the interactions of altered physical forcing (e.g. winds) and aggravated

eutrophication and it would develop into a severe hypoxic state within the next two decades. Lastly, given the significant intra-seasonal variability in low-oxygen conditions during summer, it is of great importance to conduct more cruise surveys to collect estuary-wide observations on a longer time scale in order to fully capture the generation and development of hypoxia and to confirm the change in the timing of its arrival.

*Data availability.* The marine water quality and the sediment data during 1994-2018 from the HKEPD are available at https://www.epd.gov.hk/epd/epic/english/epichome.html, while the daily wind observation data from Waglan Island automatic weather station are available at https://www.hko.gov.hk/sc/cis/climat.htm. Daily discharge data of hydrological stations (i.e. Gaoyao, Shijiao, and Boluo) can be collected at http://www.zwswj.com/cms/webfile/waterInfo/index.html.

*Author contributions.* Under the conceptualization of JH, ZC completed the data analysis and graphic visualization. This work was supervised by JH and SL. ZC wrote the paper with contributions from all co-authors and all co-authors contributed to the review and editing of manuscript, especially for JH and BW.

*Competing interests.* The authors declare that they have no conflict of interest.


*Acknowledgements.* We would like to express gratitude to the Environmental Protection Department of Hong Kong, the Waglan Island automatic weather station of the Hong Kong Observatory and the Pearl River Water Resources Commission of the Ministry of Water Resources for sharing the monitoring data.

*Financial support.* This work was supported by the Joint Research Fund of the National Natural Science Foundation of China and Guangdong Province (U1901209).

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

**Table 1 Long-term trends in the fitted LOI on average for the selected regression cases with $R^2 > 0.6$ (baseline) and for the sensitive experiments with respect to the effects of wind speeds (b), freshwater discharges (c), water temperature (d), and surface DIN concentrations (e).**

| Cases | Mean trend of LOI ($yr^{-1}$) | Changes relative to baseline (mean±std) |
|---|---|---|
| (a) baseline | 0.15 | |
| (b) WS- detrended | 0.10 | -(39%±14%) |
| (c) flow-detrended | 0.13 | -(16%±14%) |
| (d) T-detrended | 0.17 | +(11%±9%) |
| (e) DIN-detrended | 0.07 | -(56%±10%) |


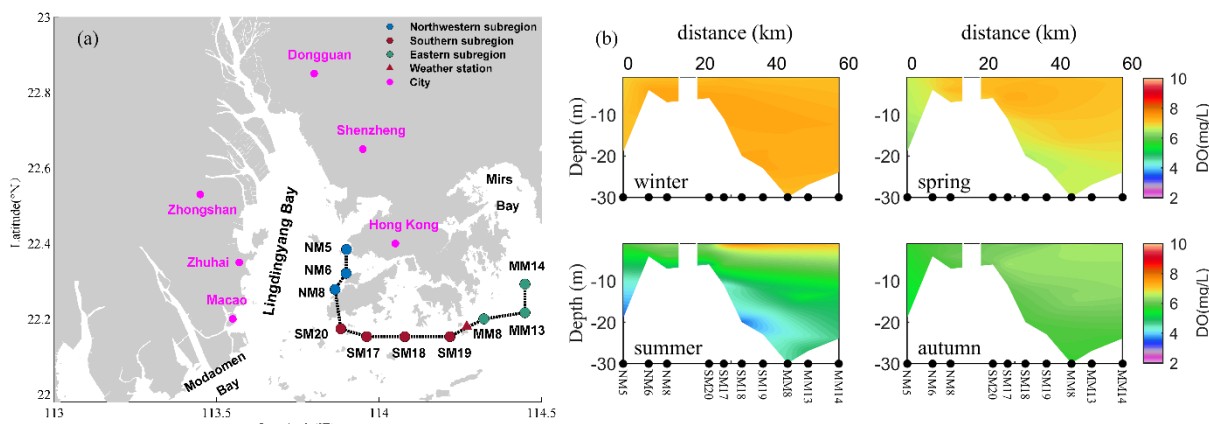

**Figure 1. (a) Map of the Pearl River Estuary (PRE) and monitoring stations in the coastal waters off Hong Kong. Note that the blue, red, green dots represent stations in the northwestern, southern, eastern subregions of Hong Kong, respectively. The red triangle denotes the location of Waglan Island automatic weather station and the purple dots indicate the location of cities in the Guangdong-Hong Kong-Macao Greater Bay Area. (b) Four subgraphs showing the vertical distributions of mean DO concentrations in winter (Dec, Jan, Feb), spring (Mar, Apr, May), summer (Jun, Jul, Aug) and autumn (Sep, Oct, Nov) during 1994-2018.**

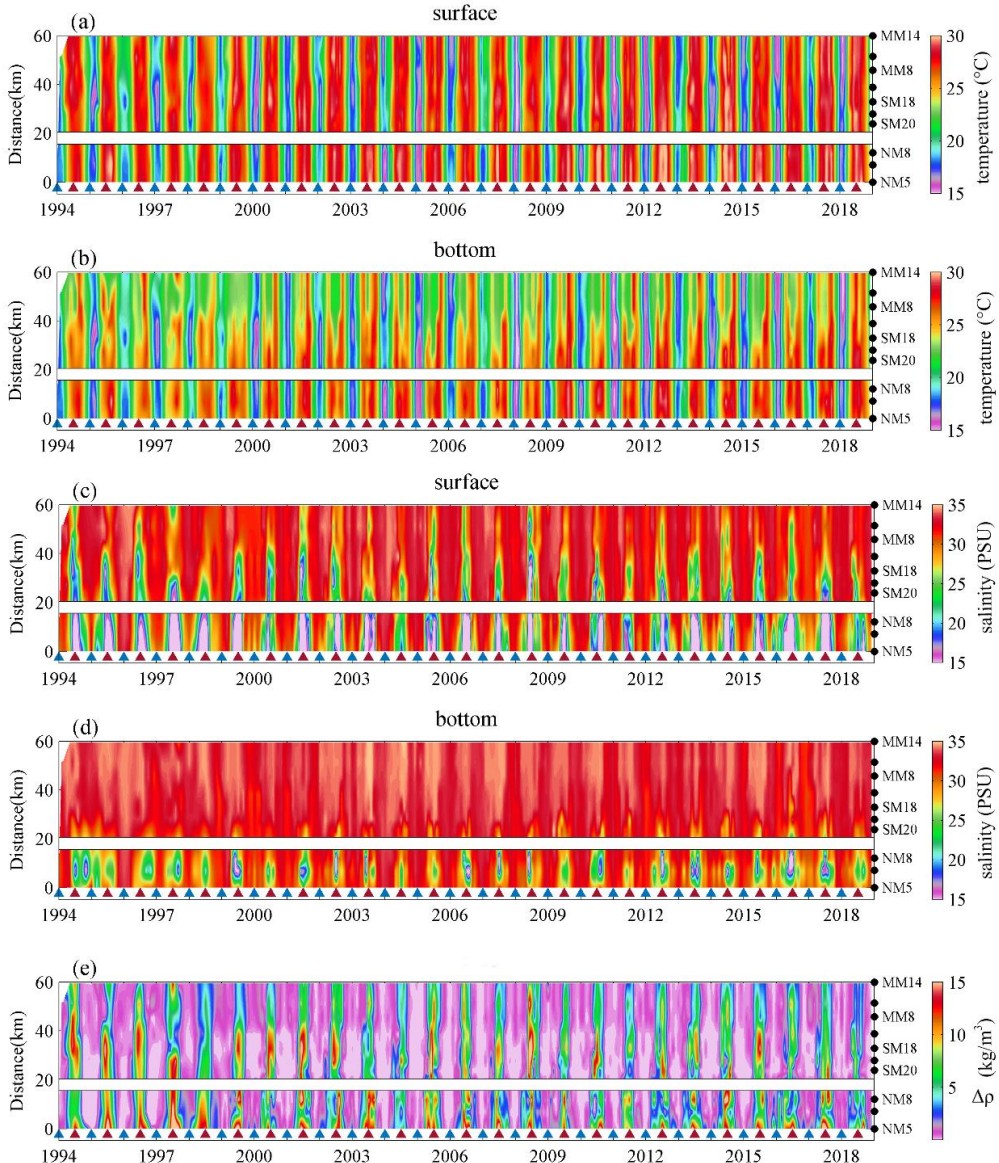


**Figure 2. Spatiotemporal distribution of temperature (a-b), salinity (c-d) in the surface and bottom waters, and vertical density differences (e) during 1994-2018. Note that the stations investigated are denoted by the black dots on the right of the figure. The blue triangles point at each December over 25 years, while the red ones point at July.**

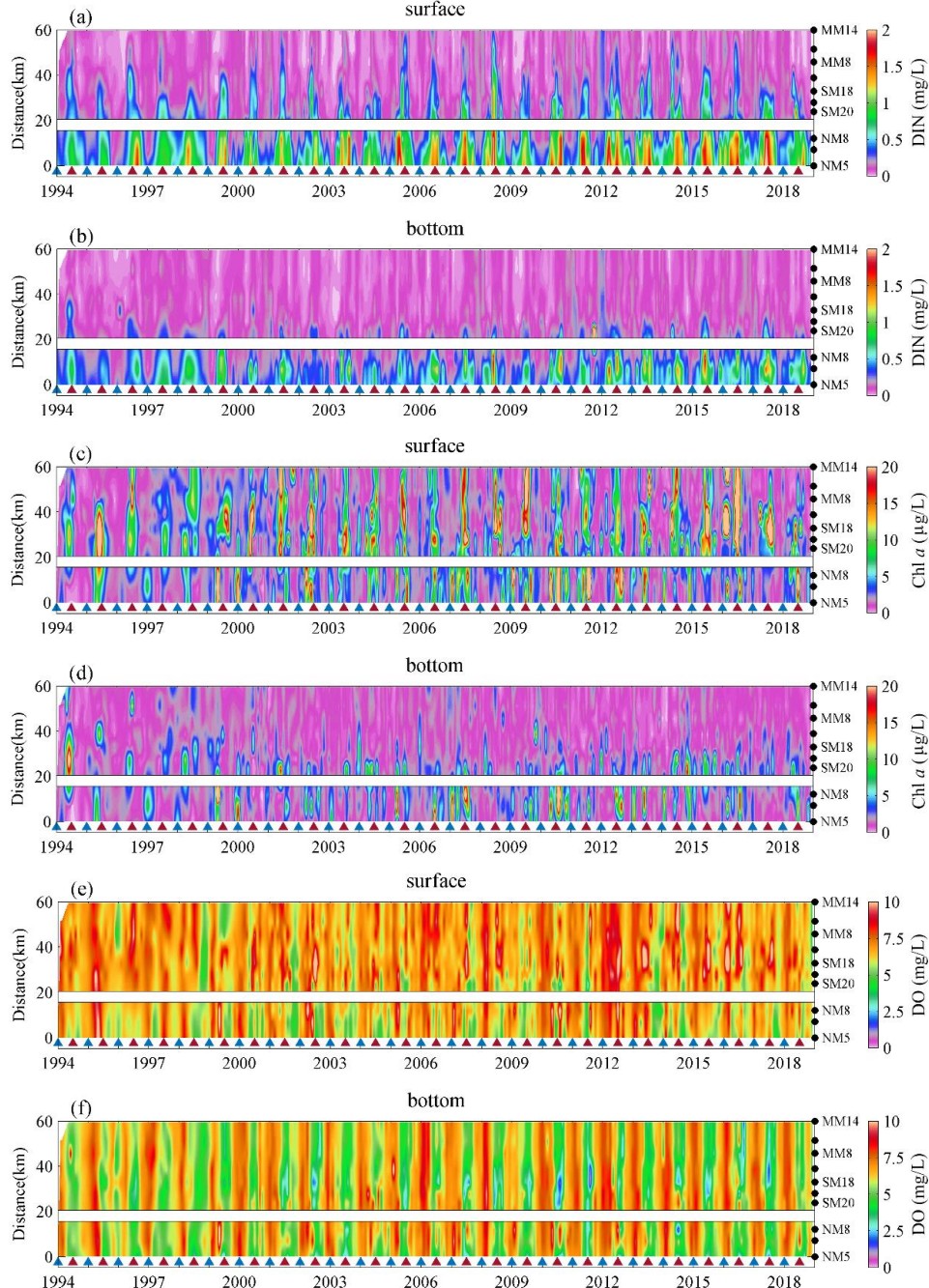

**Figure 3. Same as Figure 2 but for concentrations of DIN (a-b), Chl *a* (c-d) and DO (e-f).**

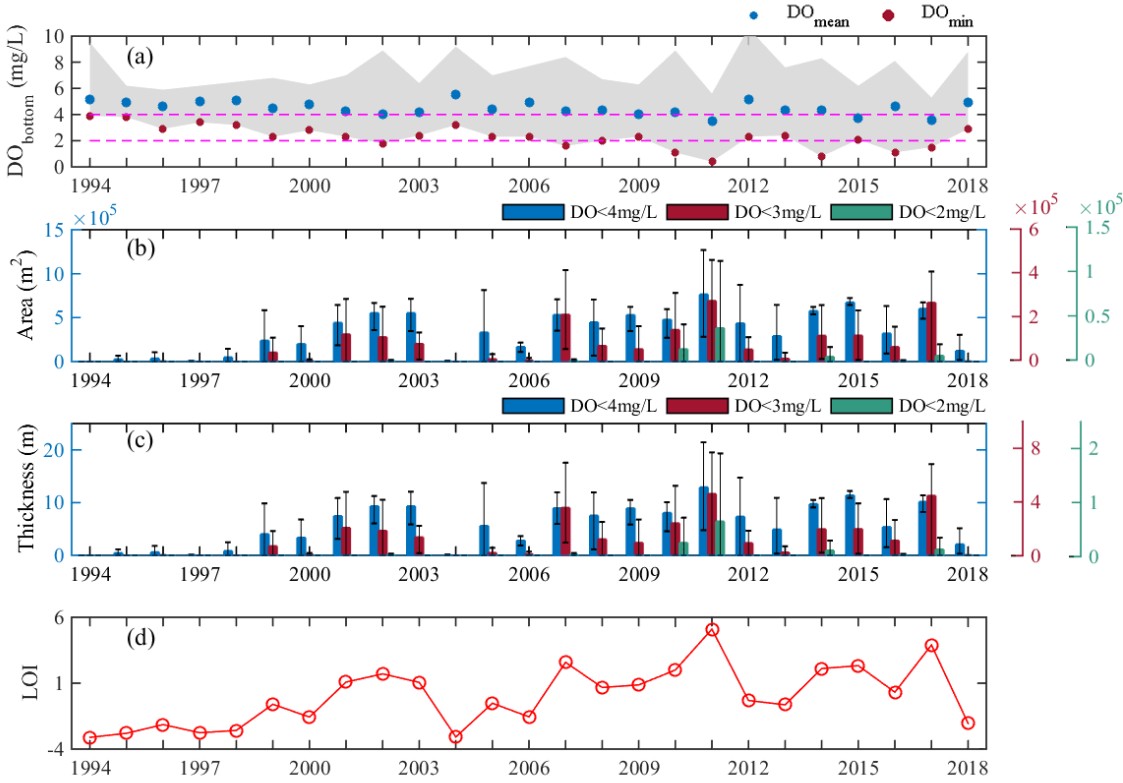

**Figure 4.** Interannual variations in the spatiotemporally (10 stations in June, July and August) mean and minimum concentrations of observed DO at the bottom (a), the cross-sectional areas (b) and layer thicknesses (c) of low-oxygen conditions, and LOI (e) in summer during 1994-2018. Note that the grey patch in (a) represents the range of bottom DO observed at the 10 stations in three summer months; the coloured bars in (b) and (c) show the mean values of three summer months, while the black thin error bars represent the range across three summer months.


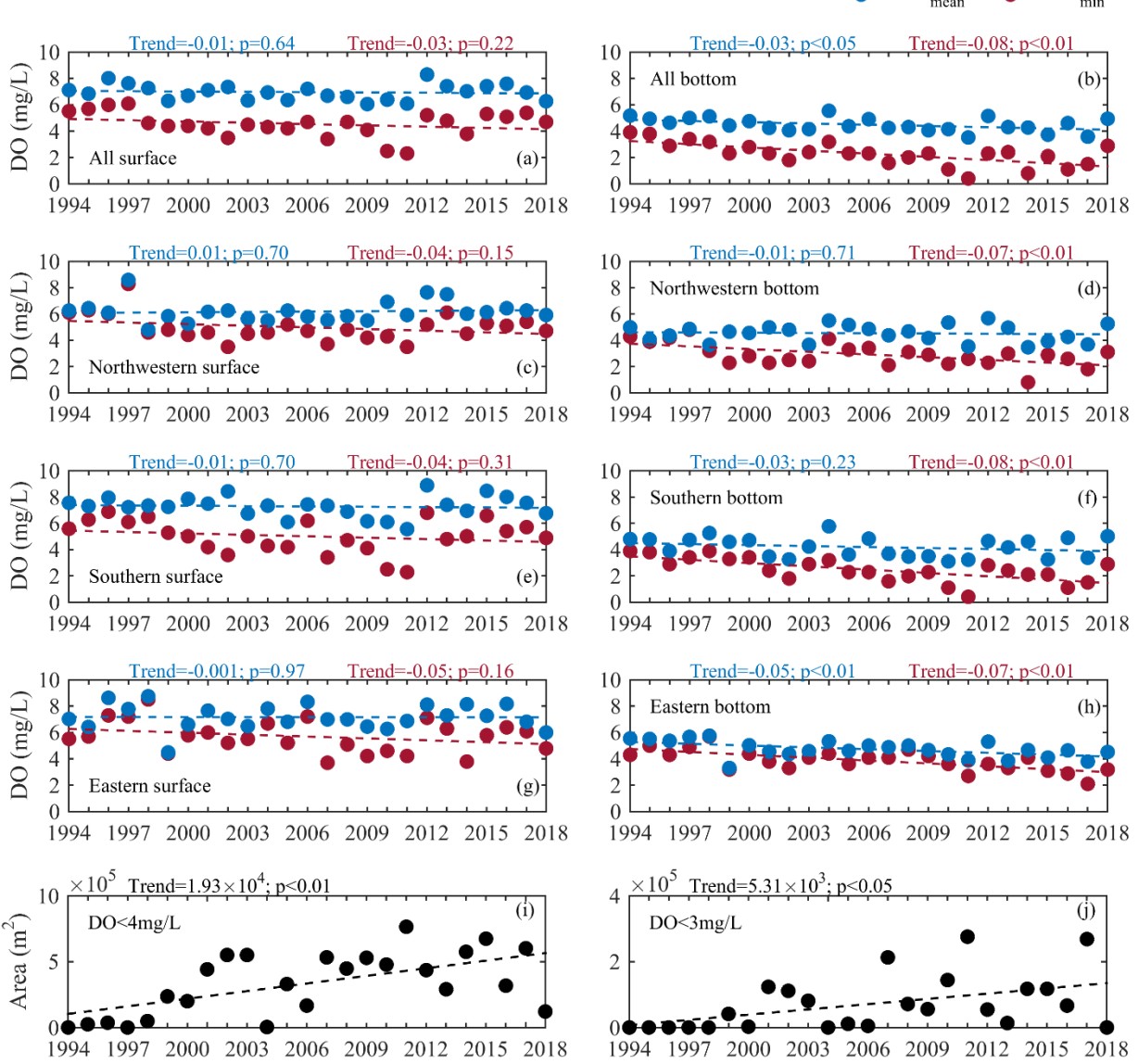

**Figure 5. Long-term trends of the mean and minimum values of observed DO at the surface and bottom waters in three summer months for all the stations (a-b) and for the northwestern (c-d), southern (e-f), and eastern (g-h) subregions, and long-term trends of the cross-sectional areas of low oxygen (i) and oxygen deficiency (j).**

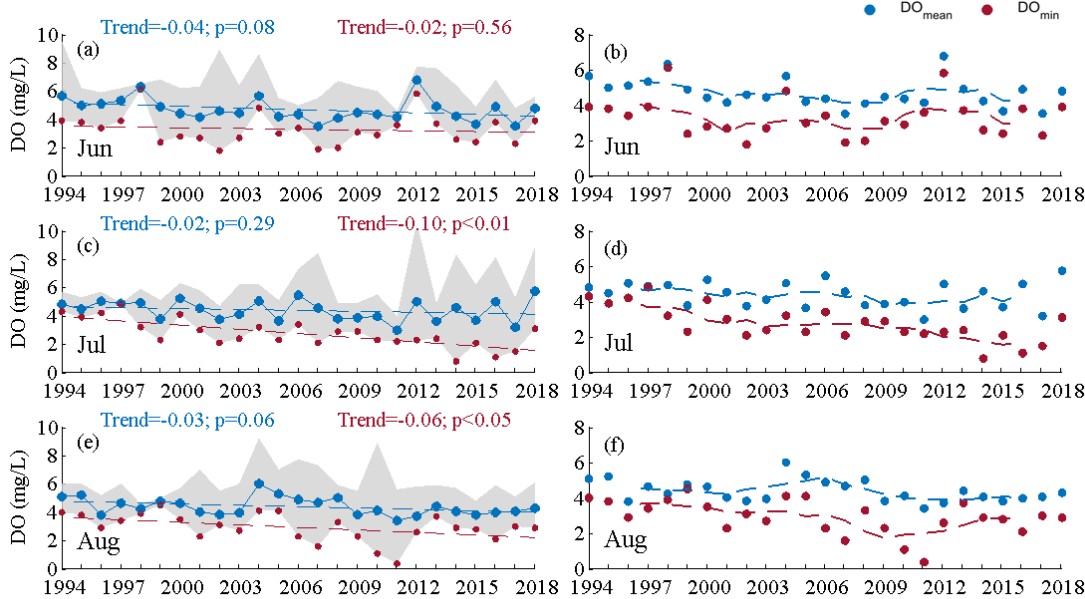

**Figure 6. Mean and minimum concentrations of DO at the bottom with their long-term trends (a, c, e) and with five-year sliding mean values (b, d, f) in summer months during 1994-2018. Note that the grey patches represent the range of DO observed in the 10 stations.**


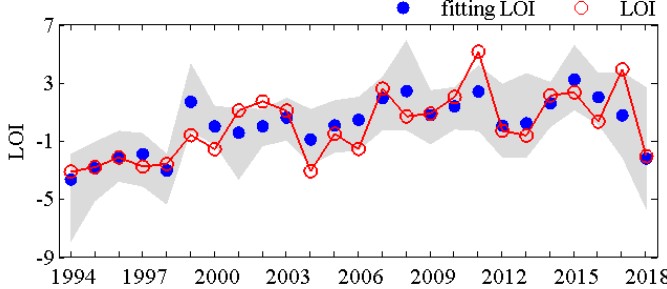

**Figure 7. Combined fitting results of the regression models with $R^2 > 0.6$ both in the training dataset and the testing dataset. Note that the red hollow dots denote the LOI estimated based on observational data, while the blue solid dots and the grey patch represent the mean values and ranges of the fitted LOI in the selected regression cases, respectively.**


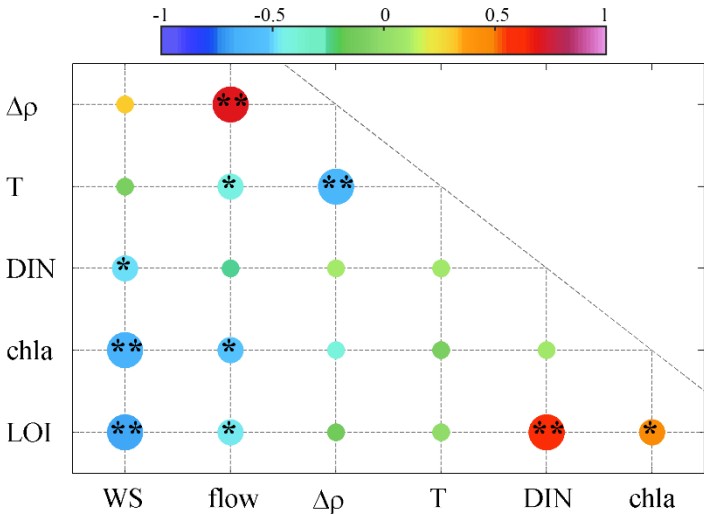

**Figure 8. Pearson correlation coefficients (r) among the wind speeds (WS), freshwater discharges (flow), vertical density differences (Δρ), bottom temperature (T), surface DIN concentrations (DIN), surface Chl *a* concentrations (Chl *a*), and LOI. Note that the color of dots show the correlation coefficients, and the symbols * and ** represents the significant level at p < 0.05 and p < 0.01, respectively.**


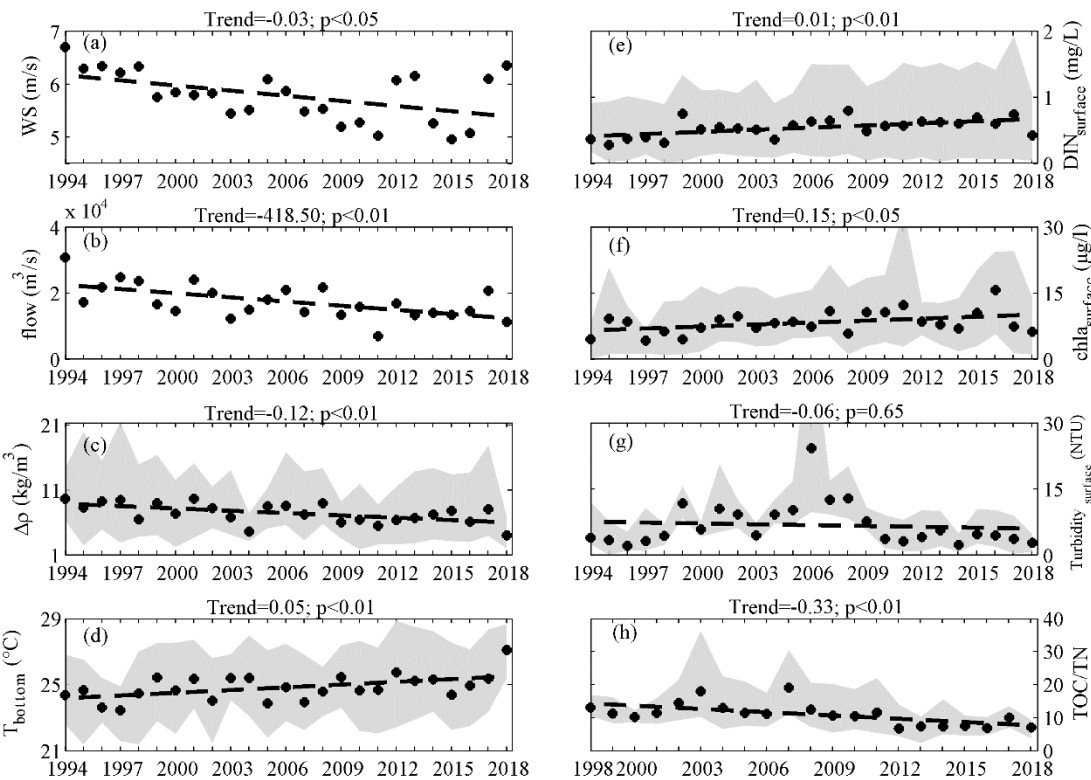

**Figure 9.** Long-term trends of the wind speeds (a), freshwater discharges (b), density differences (c), bottom temperature (d), surface DIN (e), surface Chl *a* (f), surface turbidity (g), and TOC/TN measured in the sediments (h) in summer during 1998-2018. Note that the black dots represent the spatial-average values of each variable and the grey patches represent the range of each variable observed in the 10 stations.

**Appendix**

**Table A1. Description of variables in the Principal Component Analysis (PCA)**

| Variables in PCA | Description |
|---|---|
| $DO_{mean}$ | Spatial average value of DO concentrations in bottom of each year during 1994-2018 |
| $DO_{min}$ | Spatial minimum value of DO concentrations in bottom of each year during 1994-2018 |
| $Area_4$ | Cross-sectional area of low-oxygen (DO<4 mg/L) of each year during 1994-2018 |
| $Area_3$ | Cross-sectional area of oxygen-deficiency (DO<3 mg/L) of each year during 1994-2018 |
| $Thickness_4$ | Cross-sectional thickness of low-oxygen of each year during 1994-2018 |
| $Thickness_3$ | Cross-sectional thickness of oxygen-deficiency of each year during 1994-2018 |
| Low-oxygen Index (LOI) | First principal component of PCA dimension (86.40% of variation) for measuring interannual variations in scope and intensity of oxygen conditions |


**Table A2. Total variance explained and the feature matrix of the first component in the PCA**

| Component | Eigenvalues | Variance explained (%) | Accumulative variance explained (%) | Feature matrix | Proportion |
|---|---|---|---|---|---|
| 1 | 5.184 | 86.404 | 86.404 | $DO_{mean}$ | -0.903 |
| 2 | 0.382 | 6.370 | 92.774 | $DO_{min}$ | -0.880 |
| 3 | 0.285 | 4.745 | 97.519 | $Area_4$ | 0.960 |
| 4 | 0.149 | 2.481 | 100.000 | $Area_3$ | 0.935 |
| 5 | $-1.079\times10^{-16}$ | $-1.798\times10^{-15}$ | 100.000 | $Thickness_4$ | 0.960 |
| 6 | $-4.472\times10^{-16}$ | $-7.454\times10^{-15}$ | 100.000 | $Thickness_3$ | 0.935 |

**Table A3. Coefficients of determination ($R^2$) for average wind speed (WS), southwestern wind duration (SWWD), southwestern wind cumulative stress (SWCS), southeastern wind cumulative stress (SECS) in fitting LOI; Pearson correlation coefficient of WS, SWWD, SWCS, and SECS with LOI. Note that the symbols * and ** represents the significant level at $p < 0.05$ and $p < 0.01$, respectively.**

|  | $R^2$ of fitting LOI |  | Correlation with LOI |
|---|---|---|---|
| WS+flow+T+DIN | 0.61 | WS | -0.67** |
| SWWD+flow+T+DIN | 0.55 | SWWD | 0.48* |
| SWCS+flow+T+DIN | 0.55 | SWCS | -0.33 |
| SECS+flow+T+DIN | 0.57 | SECS | 0.25 |


**Table A4. Regression coefficients of wind speed (WS), freshwater discharge (flow), bottom temperature (T) and surface DIN (DIN) of different sample datasets (Mean±Std); $R^2$ and Pearson correlation coefficient (r) of training and testing dataset in different sample datasets (Mean±Std)**

| Fitting cases | Sample size | Coefficient of WS | Coefficient of flow | Coefficient of T | Coefficient of DIN |
|---|---|---|---|---|---|
| $R^2_{train} \geq 0.6$ & $R^2_{test} \geq 0.6$ | 56010 | -0.39±0.12 | -0.14±0.12 | -0.11±0.08 | 0.49±0.12 |
| $R^2_{train} < 0.6$ or $R^2_{test} < 0.6$ | 424690 | -0.37±0.14 | -0.17±0.17 | -0.12±0.11 | 0.44±0.15 |
| Total samples | 480700 | -0.37±0.14 | -0.16±0.16 | -0.12±0.11 | 0.45±0.14 |
| | p value | $R^2_{train}$ | $R^2_{test}$ | $r_{train}$ | $r_{test}$ |
| $R^2_{train} \geq 0.6$ & $R^2_{test} \geq 0.6$ | 0.008±0.003 | 0.64±0.08 | 0.70±0.08 | 0.80±0.02 | 0.83±0.05 |
| $R^2_{train} < 0.6$ or $R^2_{test} < 0.6$ | 0.012±0.014 | 0.64±0.24 | 0.45±0.24 | 0.80±0.05 | 0.63±0.24 |
| Total samples | 0.011±0.013 | 0.64±0.24 | 0.48±0.24 | 0.80±0.05 | 0.65±0.24 |


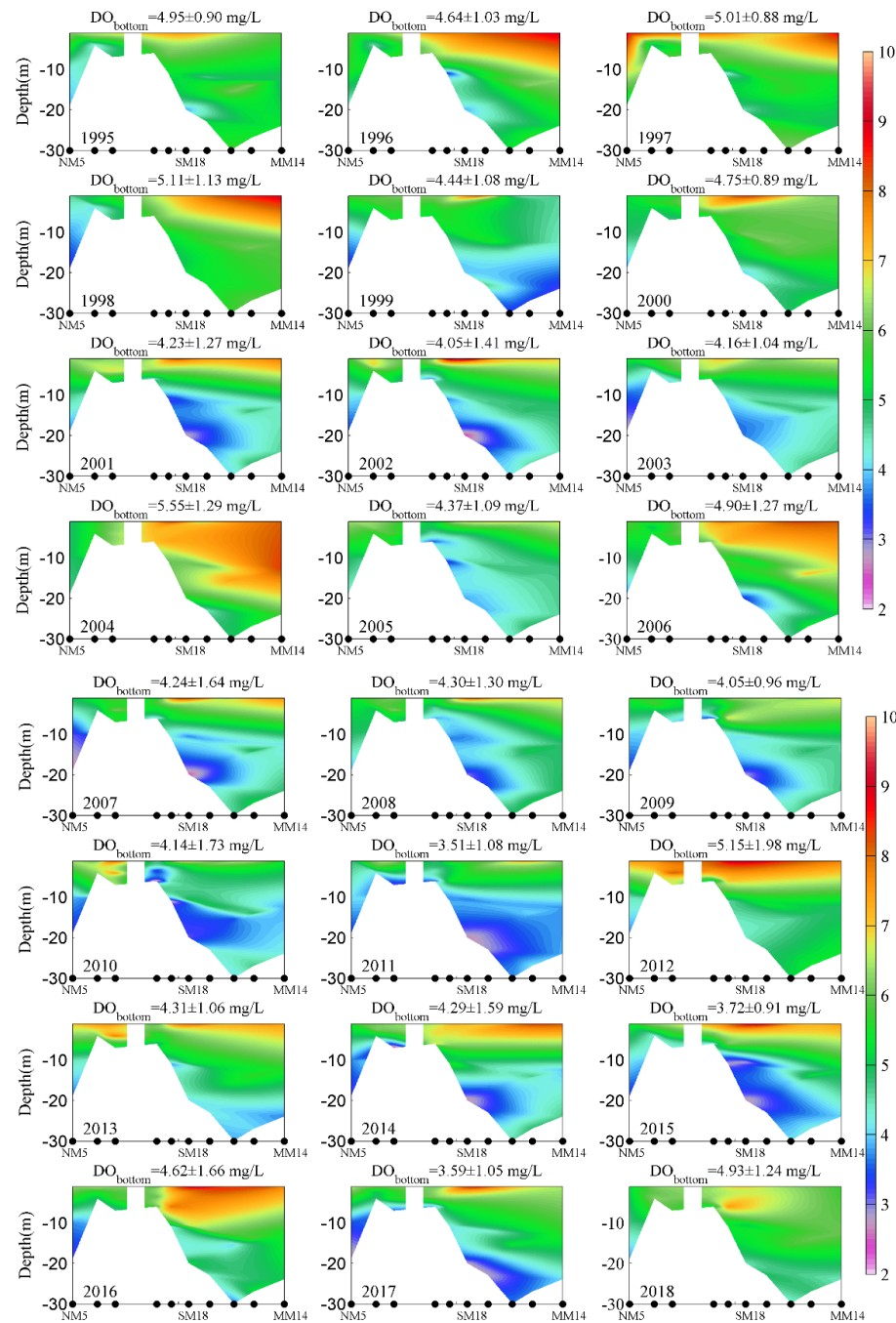

**Figure A1. Vertical distributions of average DO of all summer cruises (three months). Note that the mean values and standard deviations of bottom-water DO were also shown at the top of each subplot.**

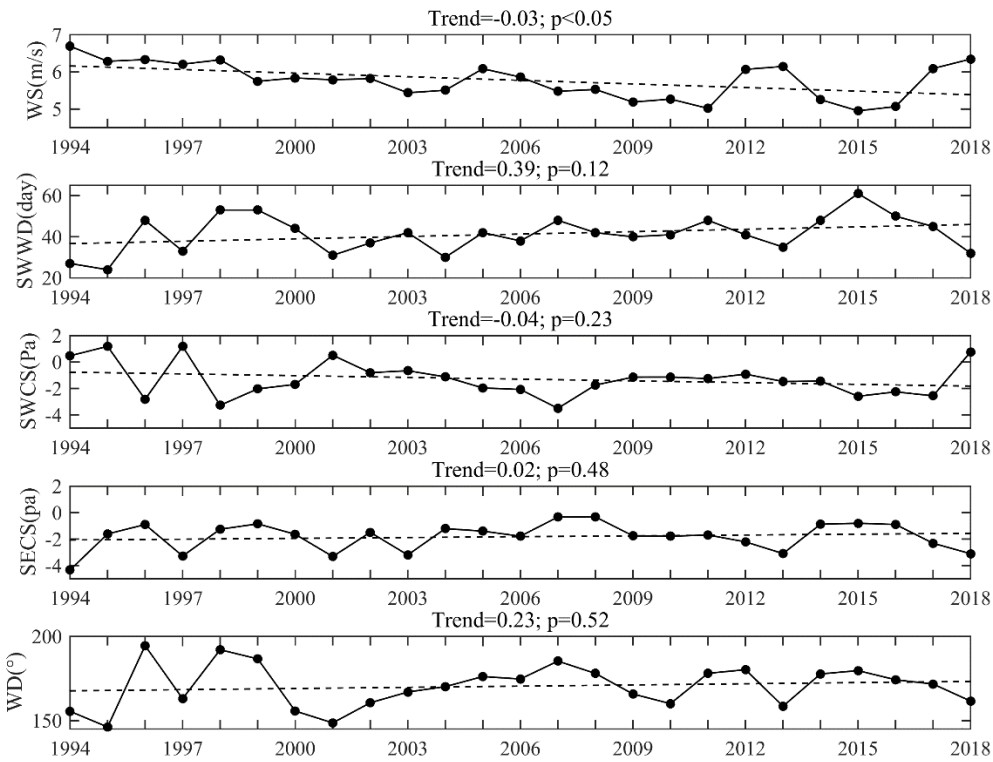

**Figure A2. Average wind speed (WS), southwestern wind duration (SWWD), southwestern wind cumulative stress (SWCS), southeastern wind cumulative stress (SECS), average wind direction (WD) in summer, and their long-term trends during 1994-2018. Note that the negative values of SWCS and SECS represent southwestern and southeastern wind, respectively. The trends and significant p values were shown in the title of each subgraph.**

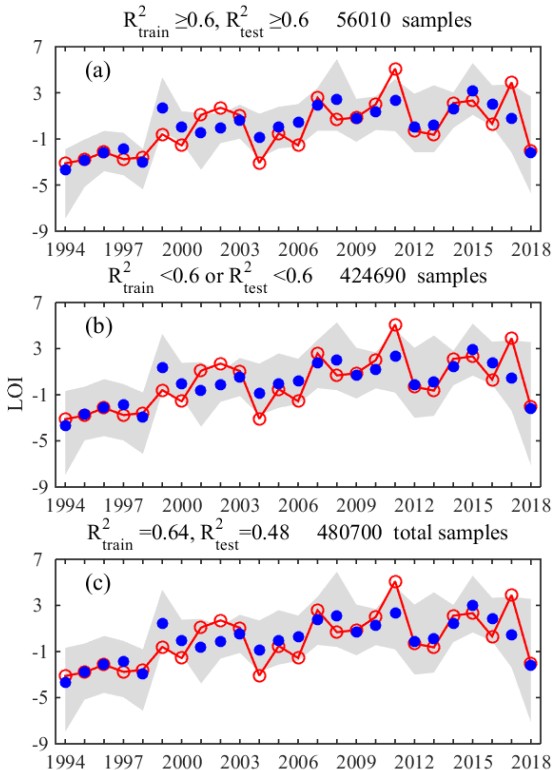

**Figure A3. Combined fitting results of the regression models with different combinations of training and testing datasets. R2 in (a) were greater than or equal to 0.6 both in training and testing datasets. R2 in (b) were less than 0.6 both in training and testing datasets. Fitting results of total samples were in (c). Note that the red hollow dots denote the LOI estimated based on observational data, while the blue solid dots and the gray patch represent the mean values and ranges of the fitted LOI in the selected regression cases, respectively.**


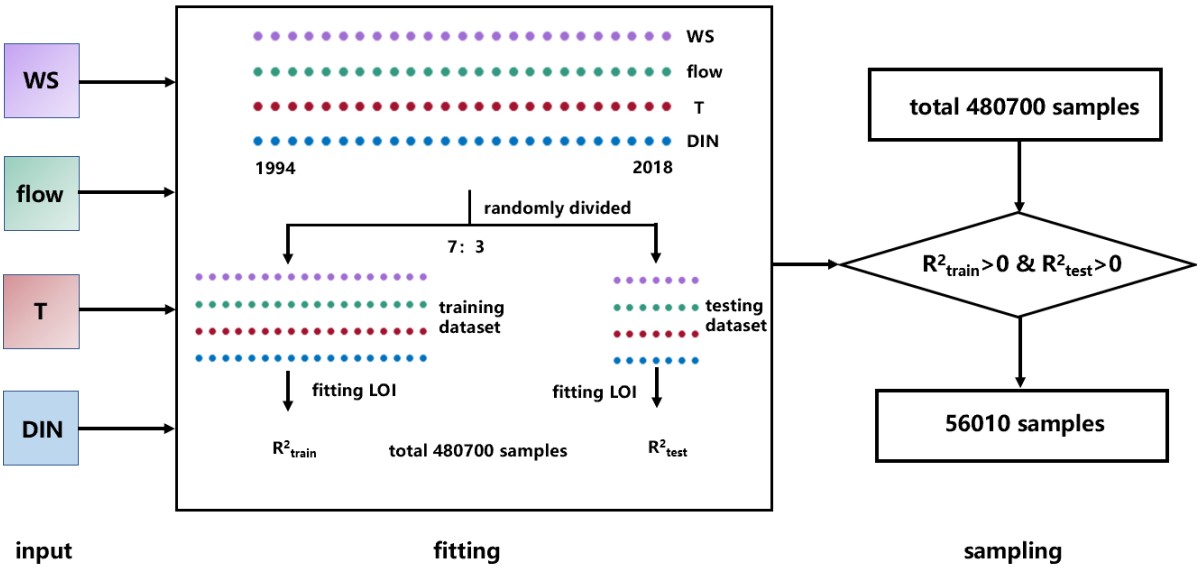


**Figure A4. Flowchart describing the fitting of LOI and the cases sampling used for analysis.**

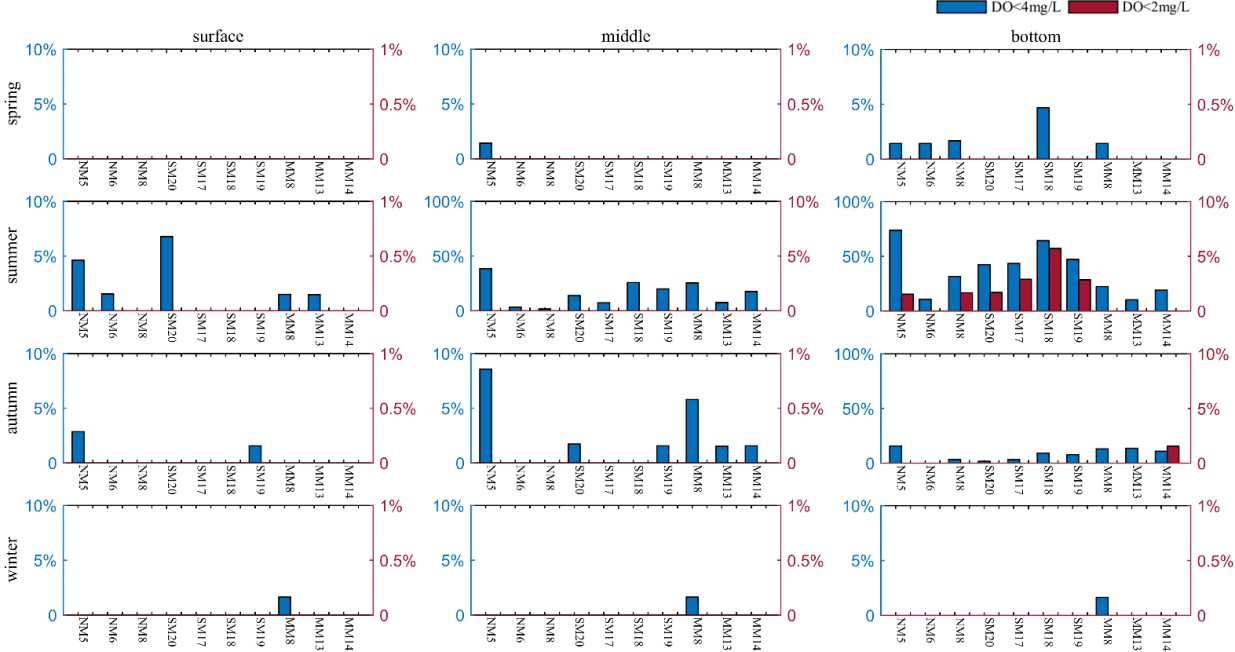

**Figure A5. Frequencies of occurrence of low-oxygen and hypoxic events in four seasons at the surface, middle, and bottom layer.**