# Peer review of "Interannual variabilities, long-term trends, and regulating factors of low-oxygen conditions in the coastal waters off Hong Kong"

_Biogeosciences, 2021_

## Author Comment (AC1)

We wish to thank the editor and referees for the constructive comments and suggestions which are helpful to the revision of our manuscript. Detailed response to all comments are given below (responses are shown in blue below the questions).

**Anonymous Referee #1**

**General Comments**

Chen et al. evaluated long-term patterns in DO in the eastern Pearl River Estuary (PRE) across seasons and regions, computed an aggregated metric of low DO, and then tested possible controlling factors of it with multiple regression. They found dissolved nitrogen and wind speed were the most explanatory variables for interannual variations and long-term trends. They use additional water quality observations to evaluate the changes to the system over time and hypothesize shifts in the system dynamics. Overall, this is a very interesting study making good use of a long-term data set to evaluate long-term change. I appreciate their thorough treatment of the data both spatially and temporally. My major comments involve clarifying the methods and what is represented in some of the graphics. Clarification is needed throughout as to which months of data are included in different average results and how the data is aggregated to represent "summer". In addition, more clarification is needed on the PCA approach as well as some re-organization of which information is presented in the Methods or Results.

Response: We are grateful to the reviewer for the positive comments and recognition of our work. We will revise the manuscript as suggested (please see details in our responses below).

**Specific Comments:**

1. **Lines 60 – 81:** Within this section, please incorporate the reasoning for your focus on the Eastern PRE. Can you describe whether this region was selected from the larger PRE because this is where the longest term data is, or is it because this is where the lowest oxygen occurs? It would provide more context if you included some description in the Introduction about how water quality in this eastern region compares to the rest of the estuary.

Response: Thank you for your comment. As Hu et al. (2021) pointed out, observational data on DO and other water quality parameters in the PRE were generally scarce in time and space, while the sampling time periods and sites and sometimes the water quality measurement methods were quite different between available datasets. These limitations and uncertainties inherent in the historical data (including the observations during 1976-2017 compiled by Hu et al. 2021) brought out great obstacles for quantifying the long-term deoxygenation trend in the PRE. By comparison, the monthly data used in our present study, collected in the coastal waters off Hong Kong, have significant merits in terms of temporal coverage (over three decades) and consistency of sampling

locations. Therefore, these data with good spatiotemporal continuity enabled us to better estimate the long-term and interannual variations in low-oxygen conditions without concerns on the uncertainties that would arise from the usage of different data sources. Besides, various oxygen-related parameters (e.g., chlorophyll-*a* concentrations) were measured as well and could be used to discern the key factors controlling the long-term oxygen changes. More importantly, the sampling sites in use were mostly located in the eastern side of the PRE, where prominent low-oxygen conditions at the bottom together with surface phytoplankton blooms have been frequently reported (Li et al., 2021; Li et al., 2020; Qian et al., 2018; Lu et al., 2018). Observational data from these coastal sites off Hong Kong (e.g. station SM18), which were close to a hotspot area of low-oxygen conditions in the eastern PRE (Hu et al., 2021), were often adopted as a representative to depict the water quality and oxygen conditions in the region (e.g. *see* Qian et al., 2018). Collectively, we considered that by utilizing the valuable dataset from Hong Kong waters, our study could provide a good insight into the long-term oxygen changes and the underlying drivers in the eastern PRE (especially from a quantitative perspective) and could be a significant part of low-oxygen researches for the whole estuary.

Based on the reviewer's suggestion, we will provide more descriptions for selection of the dataset and water quality conditions in the eastern PRE in our revised manuscript as follows:

*"... Nevertheless, due to the scarcity of observations in both time and space, a clear understanding of the long-term trend and interannual changes in hypoxia in the PRE as well as the associated drivers is still lacking, especially from a quantitative perspective. However, the spatiotemporally continuity of observational data, collected by the Environmental Protection Department of Hong Kong (HKEPD) in the eastern waters of the PRE, allowed us for a more accurate understanding of interannual and long-term low-oxygen conditions without concerns on data quality control and comparability due to uncertainties of difference dataset sources. Although the level of nutrient and terrestrial organic matter was relative lower than in the western PRE (e.g. Modaomen Bay)(Chen et al., 2020b; Yu et al., 2020), the eastern PRE was also a hotspot area of low-oxygen due to joint effects of physical and biogeochemical conditions (Li et al., 2021; Lu et al., 2018), in which research within this region could be an significant part of low-oxygen researches for the whole estuary.*

*In this study, we perform a quantitative analysis on the long-term oxygen changes (trend and interannual variability) by utilizing observational data from HKEPD..."*

2. **Line 104:** Please describe what spatial interpolation approach was used in MATLAB for the interpolations. Also, since you have land in between some of the stations, how did the method deal with that? It would be helpful to show what the region looks like in vertical cross-section as a 2nd panel of Figure 1 with the sample locations and depths indicated with dots. This would be like one of the panels of Figure A2, showing which depths each station is sampled at. This would be a helpful way to visualize the depths at each station.

Response: We estimated the vertical distribution of each water quality variable by using the "Natural-Neighbor" method, which has been widely applied in geophysics studies for its high spatial autocorrelation, in MATLAB. Regarding the island between stations NM8 and SM20, we first performed the spatial interpolations directly with all the observed data and then masked out the area cover by the island, which was roughly estimated based on its size (please note that the topographic data of the island was not available). Such a treatment has little influence on the estimation of vertical low-oxygen areas because low-oxygen conditions were seldom found in stations NM8 and SM20. Moreover, same treatment procedure was applied in each year to make it consistent when investigating the interannual variations in low-oxygen conditions.

As suggested by the reviewer, we will update Figure 1 by adding a subgraph to show the location and depth of each station.

3. **Lines 108-120:** This discussion of the PCA needs modification. Please include a table of the variables used in the PCA. I kept having to look back in the text to see how "low oxygen", "Area3," etc, were defined. I'd suggest including just that table and a description of the approach here in the Methods section. The resulting equation (Line 117) and description of it should probably be moved to the Results section. Also, please summarize the rest of the PCA results (in an appendix table), such as what % of variance the other components had, and what their weights were.

Response: Thank you for the suggestion. As suggested, we will add a table of the input variables used in the PCA (please see Table r1 below) and move Equation (1) and its associated description to the Results section (section 3.2) in our revised manuscript. In addition, we will provide more details on the rest of the PCA results in Appendix as suggested.

Table r1 Description of variables in PCA analysis

| Variables in PCA | Description |
| --- | --- |
| $DO_{mean}$ | Spatial average value of DO in bottom of each year during 1994-2018 |
| $DO_{min}$ | Spatial minimum value of DO in bottom of each year during 1994-2018 |
| $Area_4$ | Low-oxygen cross-sectional area (DO<4 mg/L) of each year of each year during 1994-2018 |
| $Area_3$ | Oxygen-deficiency cross-sectional area (DO<3 mg/L) of each year of each year during 1994-2018 |
| $Thickness_4$ | Low-oxygen cross-sectional thickness of each year of each year during 1994-2018 |

| | |
|---|---|
| Thickness$_3$ | Oxygen-deficiency cross-sectional thickness of each year of each year during 1994-2018 |
| Low-oxygen Index (LOI) | First principal component of PCA dimension (86.40% of variation), measuring interannual variations in scope and intensity of oxygen conditions |

4. **Line 125:** show an equation to describe this standardization

Response: We will show an equation to describe this standardization it as suggested.

5. **Lines 123-134:** There need to be some discussion of these different test results in the Results section.

 Response: As suggested, we will provide more details on different test results of regressions (please see Figure r1 and Table r2 below). It can be seen from Figure r1 that the ensemble means of fitting Low-oxygen Index (LOI) for different combinations of training and testing datasets were similar, but the variance explained by the fitting and the regression coefficients were different (please see Table r2). For the regression models with lower coefficients of determination ($R^2$) (Figure r1c), the fitting LOI both in training dataset and testing dataset varied in a relatively larger range. To provide a more robust data fitting result, we chose the cases with excellent performances (indicated by $R^2$ over 0.6 both in the training and testing datasets) for formal analysis. Based on the reviewer's suggestion, we will add the above discussion on different test results in section 4.1 in our revised manuscript and provide Figure r1 and Table r2 in Appendix.

[Figure]

Figure r1. Combined fitting results of the regression models with different combinations of training and testing datasets. $R^2$ in (a) were greater than or equal to 0.6 both in training and testing datasets. $R^2$ in (b) were less than 0.6 both in training and testing datasets. Fitting results of total samples were in (c). Note that the red hollow dots denote the LOI estimated based on observational data, while the blue solid dots and the gray patch represent the mean values and ranges of the fitted LOI in the selected regression cases, respectively.

Table r2 Regression coefficients of different sample datasets (Mean±Std)

| Fitting cases | WS | flow | T | DIN |
|---|---|---|---|---|
| $R_{train} \geq 0.6, R_{test} \geq 0.6$ | -0.39±0.12 | -0.14±0.12 | -0.11±0.08 | 0.49±0.12 |
| $R_{train} < 0.6$ or $R_{test} < 0.6$ | -0.37±0.14 | -0.17±0.17 | -0.12±0.11 | 0.44±0.15 |
| Total samples | -0.37±0.14 | -0.16±0.16 | -0.12±0.11 | 0.45±0.14 |

6. **Figure 4a:** can you describe the values plotted here more? Is the minimum, mean and range just from the bottom observations, or is it generated from the interpolation?
   Figure 4 (b) and (c)– We need information on the spatial interpolation to get the area and thickness. Also, if samples are collected every month, it is unclear what the bars in (b) and (c) represent. Are they the average of each month's spatially-aggregated values? If so, please put range bars on each bar to show the range across the summer months.  Or pick one month to

show.

Response: The mean and minimum DO values shown in Figure 4a are the spatiotemporally average and minimum DO concentrations for the bottom observations (using all the data from 10 stations in June, July, and August, i.e. 30 data points in total for each year). Also, the grey patch showed the minimum-to-maximum range of the observations in each year during 1994-2018. We will clarify this in the revised manuscript.

With respect to the areas and thickness of oxygen conditions, they were estimated by interpolation. Specifically, we calculated the areas and thicknesses in June, July, and August of each year, respectively, and then computed the corresponding summer means (results shown in Figure 4b-c) by averaging the areas and thicknesses of the three months. As suggested, we will clarify these estimations and revise the Figure 4b-c by adding range bars to show the range across the summer months (please see the revised figure below):

[Figure]

Figure r2. Interannual variations in the spatiotemporally (10 stations in June, July and August) mean and minimum concentrations of observed DO at the bottom (a), the cross-sectional areas (b) and layer thicknesses (c) of low-oxygen conditions, and LOI (e) in summer during 1994-2018. For (b) and (c), note that the color bars represent the mean value of three summer months, while black thin bar represent the range across three summer months.

7. **Figure A2:** Similarly to Figure 4, specify which month of the summer these plots are for. If

they are average of all the summer cruises, please justify that approach.

Response: As for the vertical DO distributions in Figure A2, they are averages of all the summer cruises. Specifically, we first averaged the observational data in the three summer months to obtain the summer means at the surface, middle, and bottom, respectively, and then interpolated them onto the profile grids to get the summertime vertical DO distributions (results shown in Figure A2). We will clarify this in the revised manuscript.

8. **Figure 5:** The min and mean DO symbols in legend seem switched.

Response: Thank you for the suggestion. We will correct this mistake.

9. **Figure 5:** I'd like to see the surface and bottom graphs with the same vertical scale (0 to 10).It can be confusing to have them different when they are right next to each other.

Response: As suggested, we will use the same vertical scale (0 to 10) in the surface and bottom graphs.

10. **Figure 5:** I'm unsure from the descriptions as to how the mean and minimum were calculated with multiple stations and months of the summer. Is the minimum the absolute minimum observed in that region in the summer, or an average of the lowest value across the stations? Also is the mean a spatial and temporal mean across the summer?

Response: As mentioned earlier (please see our response to Comment 6), the mean and minimum DO values shown in Figure 5 are the spatiotemporally average and minimum DO concentrations for the bottom observations (using all the data from 10 stations in June, July, and August, i.e. 30 data points in total for each year). Indeed, the minimum is the absolute DO minimum observed in the region in three summer months, and the mean is the spatiotemporal mean across the summer. We will revise the caption of Figure 5 to make it clear as follows:

*"Figure 5. Long-term trends of the spatiotemporally mean and minimum values of observed DO at the surface and bottom waters in three summer months for all the stations (a-b) and for the northwestern (c-d), southern (e-f), and eastern (g-h) subregions, and long-term trends of the cross-sectional areas of low oxygen (i) and oxygen deficiency (j)."*

11. **Figure 6:** The really high values in the range in recent years in July are worth mentioning. Is that just one location that is causing that range to increase, or is it some indication of increased variability?

Response: For the larger range of DO in July after 2012, it exhibited increased spatial variability. As shown in Figure r3, in July after 2012, high DO was constantly observed at the bottom of stations NM6, NM8 and SM20, while low DO was observed at the bottom of SM18 and SM19. We could find some explanations from the vertical distributions of Chl $a$ concentrations (please see Figure r3

below). The high Chl *a* in NM6, NM8 and SM20 revealed that phytoplankton flourished within the region. Due to shallow depth of these stations, high DO resulted from photosynthesis was observed at surface and bottom. However, stations located in the downstream (e.g. SM18 and SM19) would receive much organic matter transported by hydrodynamic processes. Driven by stronger water-column stratification therein, low-oxygen conditions often occurred at bottom in SM stations. After 2012, high Chl *a* occurred in July with more frequency and larger intensity, which could be main reason for larger range of DO at bottom. We will provide some discussion on this issue in our revised manuscript.

[Figure]

Figure r3. vertical distributions of DO and Chl *a* in July and August during 2012-2018.

12. **Line 233:** A diagram or flow-chart that describes the sampling and cases used in the regression analysis to get to the results would help my understanding (and probably other readers) of the methods. This could go in the Appendix.

Response: Thank you for the comment. We will add a flow-chart to describe the sampling and cases selection processes.

13. **Figure 9:** The wind speed decrease over time seems very large. Because the results indicate this is an important variable, this deserves more discussion or investigation. If the authors

already know other work that has investigated decreasing wind speeds, please cite it and describe briefly. But if there is no other research explaining this wind decrease, it would be a good idea to double-check the data and be sure that it is not an artifact of sampling dates or density shifting or sensor height changing.

Response: We totally understood the concern of the reviewer. In fact, we could find supports on the decrease of wind speed both in the Pearl River Basin (Zhang et al., 2019) and the northern South China Sea (Gao et al., 2020). Previous studies suggested that due to the weakening of atmospheric cycle activities and the East Asian monsoon (Xu et al., 2006; Zhang et al., 2009), the annual average wind speed in the Pearl River Basin exhibited a significant declining trend (at a rate of -0.003 m/s per year) during 1960-2016 (Zhang et al., 2019). In addition, previous studies showed that the long-term increase of air temperature in the Pearl River Delta region contributed to air stability and weakened the intensity of tropical cyclone (Chen et al., 2020a), which would lead to a decline of summertime wind speed in the PRE. In the northern South China sea, the summer average wind speed decreased at a rate of 0.05 m/s per year during 2004-2020 (Gao et al., 2020), which was close to that in the eastern PRE (-0.03 m/s per year). We will add more description on the decrease in wind speed in our revised manuscript (section 4.2).

14. **Appendix Figure A1:** is important b/c it doesn't suffer from any possible aggregation or averaging bias. It might be useful to make an addition panel that shows how the bottom summer counts have changed over time – maybe make one for the first half and one for the 2nd half of the record. This could also show if there's a spatial shift.

Response: Thank you for the comment. Based on the reviewer's suggestion, we have investigated changes in the occurrence of bottom low-oxygen conditions in summer during two different periods (please see Figure r4 below) to examine if there is a spatial shift. As shown, the low-oxygen and hypoxic conditions were more severe during 2006-2018 when compared to those during 1994-2005, implying a deoxygenation pattern over time. However, there is no spatial shift found between the two periods. Stations NM5 and SM18 had the highest intensity of low-oxygen events in both periods.

[Figure]

Figure r4. Frequencies of occurrence of low-oxygen and hypoxic events in four seasons at the bottom (1994-2018, 1994-2005, 2006-2018) layer.

**Technical Corrections:**

(1) **Abstract, Line 15** – change "was" to "were"

Response: We will correct it as suggested.

(2) **Abstract, line 17** – suggest changing "through the principal component analysis" to something else. Maybe "as a result of a principal component analysis"

Response: We will revise it as suggested.

(3) **Abstract, line 25** – It is unclear what "It" refers to in this sentence. Please re-write.

Response: We will revise it as follows:

*"The deteriorating eutrophication has driven a shift in the dominant source of organic matter from terrestrial inputs to in situ primary production, which has probably led to an earlier onset of hypoxia in summer."*

(4) **Abstract, last sentence** – the phrase "in the context of" is fairly awkward. Consider re-wording this sentence to make your summary stronger.

Response: We will revise it as follows:

*"In summary, the eastern PRE has undergone considerable deterioration of low-oxygen conditions driven by substantial changes in anthropogenic eutrophication and external physical factors."*

(5) **Intro, Line 33** – suggestion you use "organisms" instead of "creature"

Response: We will revise it as suggested.

(6) **Intro, Line 43-45** – Simplify (or remove) this sentence since the next few sentences cover a lot about oxygen depletion. I'd suggest just "Terrestrial organic matter discharged to estuaries can lead to intense microbial respiration."

Response: We will revise it as suggested.

(7) **Intro, Line 54:** For the Ni et al. 2020 paper, it is important to change "ocean" to "estuary." They did not study the external impact of the Atlantic Ocean warming on the Chesapeake Bay.

Response: We will revise it as suggested.

(8) **Methods, Lines 84-93** – Who collected this data?

Response: The data were provided by the Environmental Protection Department of Hong Kong. We will make it clear in the Materials and Methods section.

(9) **Results, Line 144** – I do not think the word "varied" is correct here.

Response: We will revise the sentence to *"The summertime temperature fluctuated between 28.21 $\pm 1.19\ °C..."* . All the relevant phrases in the manuscript will be revised accordingly.

(10) **Results, Line 168** – wording like this sentence can be simplified. You could just start with "DO concentrations exhibited significant…"

Response: We will revise it as suggested.

(11) **Results, Line 192, and other places** – The phrase "DO content" is not something I've seen very much before in the hypoxia literature. I'd suggest using "DO concentrations" or just "DO".

Response: We will revise it as suggested.

(12) **Figure 6** – It would be helpful to use the same open circles for the blue symbols as in Figure 5.

Response: As suggested, we will change the legends of $DO_{mean}$ and $DO_{min}$ in Figure 6 and make them consistent with those in Figure 5.

(13) **Discussion, Lines 297-298:** Please revise the sentence that starts with "As quantified by statistic methods…" to work on the wording. Maybe "Our analysis showed that increasing DIN…"

Response: We will revise it as suggested.

**Reference**

Chen, J., Wang, Z., Tam, C.-Y., Lau, N.-C., Lau, D.-S. D., and Mok, H.-Y.: Impacts of climate change on tropical cyclones and induced storm surges in the Pearl River Delta region using pseudo-global-warming method, Scientific Reports, 10, 1965, 10.1038/s41598-020-58824-8, 2020a.

Chen, L., Zhang, X., He, B., Liu, J., Lu, Y., Liu, H., Dai, M., Gan, J., and Kao, S.-J. I.: Dark Ammonium Transformations in the Pearl River Estuary During Summer, Journal of Geophysical Research: Biogeosciences, 125, 10.1029/2019JG005596, 2020b.

Gao, N., Ma, Y., Zhao, M., Zhang, L., Zhan, H., and He, Q.: Quantile Analysis of Long-Term Trends of Near-Surface Chlorophyll-a in the Pearl River Plume, Water, 12, 1662, 10.3390/w12061662, 2020.

Hu, J., Zhang, Z., Wang, B., and Huang, J.: Long-term spatiotemporal variations and expansion of low-oxygen conditions in the Pearl River estuary: A study synthesizing observations during 1976–2017, 10.5194/bg-2020-480, 2021.

Li, D., Gan, J., Hui, C., Yu, L., Liu, Z., Lu, Z., Kao, S.-j., and Dai, M.: Spatiotemporal Development and Dissipation of Hypoxia Induced by Variable Wind-Driven Shelf Circulation off the Pearl River Estuary: Observational and Modeling Studies, Journal of Geophysical Research: Oceans, 126, e2020JC016700, https://doi.org/10.1029/2020JC016700, 2021.

Li, X., Lu, C., Zhang, Y., Zhao, H., Wang, J., Liu, H., and Yin, K.: Low dissolved oxygen in the Pearl River estuary in summer: Long-term spatio-temporal patterns, trends, and regulating factors, Marine Pollution Bulletin, 151, 110814, https://doi.org/10.1016/j.marpolbul.2019.110814, 2020.

Lu, Z., Gan, J., Dai, M., Liu, H., and Zhao, X.: Joint effects of extrinsic biophysical fluxes and intrinsic hydrodynamics on the formation of hypoxia west off the Pearl River Estuary, Journal of Geophysical Research: Oceans, 123, 6241-6259, 2018.

Qian, W., Gan, J., Liu, J., He, B., Lu, Z., Guo, X., Wang, D., Guo, L., Huang, T., and Dai, M.: Current status of emerging hypoxia in a eutrophic estuary: The lower reach of the Pearl River Estuary, China, Estuarine, Coastal and Shelf Science, 205, 58-67, https://doi.org/10.1016/j.ecss.2018.03.004, 2018.

Xu, C.-Y., Gong, L., Tong, J., Chen, D., and Singh, V.: Analysis of spatial distribution and temporal trend of reference ET in Changjiang catchments, Journal of Hydrology, 327, 81-93, 10.1016/j.jhydrol.2005.11.029, 2006.

Yu, L., Gan, J., Dai, M., Hui, C., lu, x., and Li, D.: Modeling the role of riverine organic matter in hypoxia formation within the coastal transition zone off the Pearl River Estuary, Limnology and Oceanography, 9999, 1-17, 10.1002/lno.11616, 2020.

Zhang, T., Chen, Y., and U, K.: Quantifying the impact of climate variables on reference evapotranspiration in Pearl River Basin, China, Hydrological Sciences Journal, 64, 1-13, 10.1080/02626667.2019.1662021, 2019.

Zhang, X., Ren, Y., Yin, J., Lin, Z., and Zheng, D.: Spatial and temporal variation patterns of reference evapotranspiration across the Qinghai-Tibetan Plateau during 1971–2004, J. Geophys. Res, 114, 10.1029/2009JD011753, 2009.

---

## Author Comment (AC2)

We wish to thank the editor and referees for the constructive comments and suggestions which are helpful to the revision of our manuscript. Detailed response to all comments are given below (responses are shown in blue below the questions).

**Anonymous Referee #2**

**General Comments**

1. The manuscript by Chen et al. provides data analysis (mostly statistical) of the water quality observations in the southeastern part of the Pearl River Estuary (PRE). Several recent studies, such as Hu et al., 2021, and Li et al., 2021, about the de-oxygenation problems in this region, provide background and justification of hypoxia-related study in PRE. Nevertheless, I feel this manuscript fails to connect the new data with the findings of these existing studies and covers only a small portion of the PRE; thus, its potential to convey what can be learned from a regional study to a broader audience is limited.

   A critical flaw of this study is the representativeness of the stations that all statistical analyses are built upon. I acknowledge these are valuable (30 yr) monthly data covered by these stations, yet their spatial coverage is mainly surrounding the Hongkong island; it is a challenge to draw any solid conclusion regarding the PRE, even the east part, without a throughout cross-reference with the model and data by the previously mentioned two studies. Instead of using these stations to refer to east PRE, I would say the author should do the opposite—to provide a possible water quality study of the Hongkong coast with impacts from the PRE.

   Built on the above point, I see in line 103 (page 4), "the observed DO profiles were interpolated by MATLAB along with the three subregions with a grid resolution of 600 m (length) and 0.3 m (depth)." But take a look at the distribution of the stations; they are concentrated mainly in the nearshore area surrounding Hongkong; I doubt they can be representative of the condition in the east part of the PRE (e.g., the ECTZ defined by Li et al., 2021). And any conclusion based on such "hypoxia area" analysis (e.g., Fig. 4 ) is thus questionable.

   Response: Thank you for the comments. As the reviewer pointed out, there have been many observational and modelling studies on hypoxia in the PRE over recent years. However, most of these studies focused on short-term oxygen dynamics (Li et al., 2021), whereas the long-term oxygen pattern was less investigated. Hu et al. (2021) had an attempt to elucidate the long-term evolution of low-oxygen conditions in the PRE using observations during 1976-2017 collected from multiple datasets; however, as they mentioned, there existed data gaps in certain years and lack of conformity in observational coverage, and the observations compiled were under sampled in several years, which brought out great obstacles for quantifying the long-term oxygen changes due to the data limitations (Hu et al., 2021). Given this situation, we decided to utilize the monthly data collected in the coastal waters off Hong Kong, which have significant merits in terms of temporal

coverage (over three decades) and consistency of sampling locations. These coastal data with good spatiotemporal continuity enabled us to better estimate the long-term and interannual variations in low-oxygen conditions without concerns on the uncertainties that would arise from the usage of different data sources. Besides, various oxygen-related parameters (e.g., chlorophyll-*a* concentrations) were measured as well and could be used to discern the key factors controlling the long-term oxygen changes. It should be mentioned that several previous studies have also used data from the same source (e.g. station SM18) to represent water quality conditions in the eastern PRE (Lui et al., 2020). Therefore, we considered that our data selection was reasonable.

Moreover, based on in-situ observations in the PRE, low-oxygen conditions exhibited large spatial variations. Low-oxygen conditions were observed in the upstream zone (Li et al., 2020) and western shelf (Wang et al., 2016; Yin et al., 2004; Cui et al., 2018; Shi et al., 2019). Also, in the eastern PRE, prominent low-oxygen conditions at the bottom together with surface phytoplankton blooms have been frequently reported (Li et al., 2021; Li et al., 2020; Qian et al., 2018; Lu et al., 2018). Observational data from the coastal sites off Hong Kong (e.g. station SM18), which were close to a hotspot area of low-oxygen conditions in the eastern PRE (Hu et al., 2021), were often adopted as a representative to depict the water quality and oxygen conditions in the region (e.g. Qian et al., 2018 and Lui et al., 2020). Besides, dominant deoxygenation mechanism varied among subregions in the PRE. While the western coastal transition zone (CTZ) (Li et al., 2021) was more heavily influenced by terrestrial organic matter (Wang et al., 2016; Wang et al., 2018), low-oxygen events in the Hong Kong waters were largely regulated by the joint effect of physical and eutrophic processes (Li et al., 2021; Qian et al., 2018). What's more, we found that the deterioration trend of low-oxygen in the Hong Kong waters was consistent with the whole PRE (Hu et al., 2021), revealing that the Hong Kong data could reflect some characteristics of the long-term deterioration of low-oxygen conditions in the PRE. Collectively, although the stations selected in our study couldn't cover the whole area of the eastern PRE, we considered that these stations could grasp the long-term deoxygenation trend in the low-oxygen hotpot area in the eastern PRE. In addition, by utilizing this valuable dataset from Hong Kong waters, we could also provide a better insight into the long-term oxygen changes and the underlying mechanisms in the eastern PRE from a quantitative perspective, which could be a significant part of low-oxygen researches for the whole estuary.

Based on the reviewer's comment, we will revise the manuscript by 1) adding more descriptions of water quality and oxygen background information in different subregions (including the upstream regions and the western and eastern CTZ) in section 1 to better justify the representativeness of our analysis using the Hong Kong data, and enhancing the descriptions regarding the current status and existing problems in the long-term low-oxygen research in the PRE to emphasize the significance of our work; 2) adding more discussion on the effects of wind, freshwater discharge, and nutrient load on deoxygenation in combination with the relevant findings of the previous studies, and enhancing the analysis on differences of key deoxygenation mechanism

in different subregions of the PRE to improve the implication of our work; and 3) further clarifying the selection on the external variables (e.g. wind speed and direction) in the Methods section. Overall, we hope to emphasize our aim in exploring the long-term low-oxygen change and its important drivers for a significant part of the PRE, in which our analysis methods exhibited general applicability for further studies.

[Figure]

Figure r1 Observation sites during 1976-2014 in the PRE in previous studies (Hu et al., 2021) and station sites in our work (a). Note that black dots represent observation sites in previous studies and blue, red, green dots represent station sites in the northwestern, southern, and eastern subregions of Hong Kong waters, respectively. The incidence of low-oxygen (b1) and oxygen-deficiency (b2) conditions at the bottom of the PRE in summer during 1976-2014 (Hu et al., 2021).

2. In addition, the water depth of these stations varies from 5 m to more than 30m, and the analysis in this manuscript concentrated mostly surface and bottom, which impair the reliability of such analysis in the de-oxygenation study, which is very sensitive to water depth and vertical distribution of variables.

Response: We agreed that the analysis of deoxygenation and its related processes was sensitive to water depth and vertical distribution of variables. In fact, we did concern with this issue, especially for DO. That is exactly the reason why we used several vertical-profile-based metrics (e.g. the cross-sectional area and the layer thickness of low-oxygen conditions) to depict the long-term oxygen changes in the region (please see detailed descriptions in the Methods section 2.2). For example, as for the spatial extent of low-oxygen conditions, we estimated the vertical low-oxygen area and thickness by interpolation using the data at the surface, middle, and bottom layers according to the depth of each station. In addition, we have also investigated the occurrence of low-oxygen and hypoxic events at the surface, middle, and bottom layers (please see Figure A1 in the manuscript), which indicated that low-oxygen events mainly appeared in the bottom waters of summer. Vertical distributions of DO in summer was provided as well (please see Figure A2 in the Appendix). For other variables, we also performed a thorough analysis on their spatial (surface, middle, bottom

layers) and temporal patterns. We found that the key problems were concerned mostly in the surface and bottom waters. For instance, while low DO mostly occurred at the bottom, high Chl *a* concentrations frequently occurred at the surface. Therefore, in order to embody the compactness of analysis and data display, we decided to focus on the surface and the bottom when showing part of the results on the long-term variations in water quality constituents and DO concentrations (e.g. results shown in Figures 2-3). Nevertheless, we would like to emphasize that several oxygen-related metrics incorporating information of vertical DO distributions were analyzed and discussed throughout the manuscript, including the Low-Oxygen Index (LOI as defined in section 2.2) and the cross-sectional area of low-oxygen conditions.

3.  In the abstract, the author indicated that "there is still a lack of quantitative understanding of the long-term trends and interannual variabilities in oxygen conditions in the PRE as well as the driving factors, which was comprehensively investigated in this study," which I could not agree, I think Hu et al. and Li et al. provided good studies about the mechanism of oxygen dynamics in the PRE. Yet, this manuscript fails to connect what is observed by the stations surrounding Hongkong to what has been reported in a larger geospatial content (PRE and coastal/shelf water).

Response: Thank you for your comments. For the sentence *"there is still a lack of quantitative understanding of the long-term trends and interannual variabilities in oxygen conditions in the PRE as well as the driving factors, which was comprehensively investigated in this study"* that we mentioned in manuscript, we would like to provide some explanations. As we mentioned earlier (please see our response to Comment 1), there were indeed many studies about the mechanism of hypoxic generation and development in the PRE, but most of them focused on short-term oxygen dynamics (Li et al., 2021), while the long-term mechanism received less attention. Hu et al. (2021) had an attempt to elucidate the long-term evolution of low-oxygen conditions. However, as mentioned in their analysis, more quantitative perspective on hypoxic mechanism was needed, limited by the lack of accessible continuous observations. Therefore, the long-term analysis of Hong Kong data could make up for these problems to some extent as mentioned earlier (please see our response to Comment 1).

With respect to the connection of our analysis with previous studies, we also believed that it was vitally necessary. Although most of researches of low-oxygen focused on the whole PRE, we tried to compare their results involved in the eastern CTZ with ours (please see section 4.1 and 4.2). For example, as for the effect of wind (in section 4.1), we have explained our results by combining the findings of previous studies (please see details in Line 55-58). We agreed that this part of discussion should be enhanced. As we mentioned earlier (please see our response to Comment 1), we will provide more detailed discussions in this part. For instance, in section 4.1, we will improve the reference on the effect of wind on the circulation and material fluxes (Li et al., 2021) in the

eastern PRE and make clearer explanation on our focus on wind speed. Also, we will add more comparison about key low-oxygen driving factors among different subregions of the PRE in our revised manuscript.

4. The author uses wind speed in their statistic analysis which is also questionable.  How about wind direction, and does the wind play the same role in low oxygen development over a year? I am asking because Li et al. (2021) indicated that both wind direction and intensity influenced the circulation nutrient flux, detritus, and vertical mixing. Also, as suggested by Li et al., 2021, what is the role of shelf circulation in physics (mixing, etc.) and nutrient and sediment delivery? Li et al. (2021) and Feng et al. (2014) show that the upwelling and downwelling favorable wind condition has different impacts on the low-oxygen development in this area. It is problematic to use the monthly mean wind speed as a predictor without looking into wind's detailed role in this environment.

Response: In fact, during the preparation of our study, we have also tried to examine the effect of wind direction since the transport of nutrient and detritus, vertical mixing, and residence time could be regulated by wind-driven circulation (Li et al., 2021). Previous studies have revealed that the southwestward (upwelling-favorable) wind could enhance hypoxia in the eastern PRE (Li et al., 2021). Therefore, we processed the daily data of wind speed and direction into monthly average wind speed (WS), southwestward wind duration (SWWD), southwestward wind cumulative stress (SWCS), and southeastward wind cumulative stress (SECS) in summer (please see Figure r2 below), and then we carried out a suite of multiple regressions to fit the Low-oxygen Index (LOI) for each wind-related variable. As shown in Table r1 below, the fitting effect of LOI was better using WS, which also has the highest correlation with LOI among the wind-related variables. We considered that WS was the most significant explanatory wind-related factor for the interannual variations in LOI, and thus we eventually selected WS to be the wind-related input variable in the multiple regression. Besides, we noted that the long-term trends of direction-related variables (i.e. SWWD, SWCS, and SECS) were not as significant as WS (please see Figure r2 below). This probably suggested that the wind direction-related processes played vital important roles in short-term low-oxygen events, but the decline in wind speed has a more significant contribution from a long-term perspective. Based on the reviewer's concern on the wind direction, we will provide more discussion (as mentioned above) on this issue in our revised manuscript.

[Figure]

Figure r2. Average wind speed (WS), Southwestward wind duration (SWWD), Southwestward wind cumulative stress (SWCS), Southeastward wind cumulative stress (SECS) in summer and their long-term trends during 1994-2018. Note that the negative value of SWCS and SECS represent southwestward and southeastward wind respectively. The trend and significant p value were texted on the top of each subgraph.

Table r1 $R^2$ of Average wind speed (WS), Southwestward wind duration (SWWD), Southwestward wind cumulative stress (SWCS), Southeastward wind cumulative stress (SECS) participating in fitting LOI; Pearson correlation coefficient of WS, SWWD, SWCS, SECS with LOI. Note that the symbols * and ** represents the significant level at $p < 0.05$ and $p < 0.01$, respectively.

|  | $R^2$ of fitting LOI |  | Correlation with LOI |
| --- | --- | --- | --- |
| WS+flow+T+DIN | 0.61 | WS | -0.67** |
| SWWD+flow+T+DIN | 0.55 | SWWD | 0.48* |
| SWCS+flow+T+DIN | 0.55 | SWCS | -0.33 |
| SECS+flow+T+DIN | 0.57 | SECS | 0.25 |

5. The author's conclusion that the eastern PRE would "develop into a severe hypoxic state within the next two decades" is too strong to be supported by the analysis provided by this study. For

instance, what do the wind, large-scale circulation (cause it affects lateral delivery of water and nutrient, etc.), and river (Pearl plus wastewater from the city) look like in the next two decades? The Pearl River discharged into the PRE from the north, yet if we focus on the spatial scale covered by the stations in this study, what is Pearl River's role in MM stations?    Also, what is the impact of overland runoff from Hongkong, such as wastewater discharge, which is also indicated by Hu et al. (2021), and the author briefly mentioned this in Line 250 of page 8.

Response: Based on the observational data and the multiple regression model, we have discerned the contributions of wind speed, freshwater discharge, water temperature, and DIN to the long-term deoxygenation, in which wind speed and DIN contributed the most. Based on the wind data, the wind speed was expected to decrease at a rate of 0.6 m/s within two decades, which would contribute to longer water residence time and nutrient accumulation (Li et al., 2021) in combination with wind-driven circulation, favoring the generation and development of low-oxygen conditions. Likewise, DIN was expected to increase by 0.2 mg/L in two decades according to the observed growth trend, which would enhance eutrophication and subsequent low-oxygen condition within the region. Overall, based on the long-term trends of each influential variable and the proposed regression model, we made the judgement that low-oxygen conditions would develop into a severe hypoxic state. Certainly, this inference was laid on the assumption that wind speed, freshwater discharge, water temperature, and DIN would keep increasing/decreasing at the rate in the next two decades.

We totally understood the reviewer's concern. It should be mentioned that the long-term changes in low-oxygen conditions were ultimately driven by the long-term changes in physical factors (e.g. wind and circulation) and biogeochemical factors (e.g. nutrients), and the proposed regression model (fitting the LOI) actually implicitly incorporated the influence of physical-biogeochemical processes on DO to some extent. We have also tried to discuss the effects of wind, freshwater discharge, and DIN in combination with the circulation and nutrient fluxes in section 4.1 and 4.2, but we realized that it was hard to explore the detailed changes in mechanic processes clearly based on observational data and statistic methods only. To this end, further studies such as numerical simulation are needed in the future. Also, we would like to point out that although prediction with the multiple regression would bring out uncertainties by losing some non-linear physical-biological processes, the long-term deoxygenation trends revealed by the regression model could provide implication for scientific research and management of low-oxygen conditions in the region. Based on the reviewer's comment, we will improve this part of discussion in our revised manuscript. Specifically, in section 5 we will add the rationale of our prediction on the deterioration of low-oxygen conditions. Moreover, we will enhance the discussion on the combined effects of mechanic processes as mentioned above in the revised manuscript, and we will also discuss the limitation of our statistic methods to have a clearer implication of our work.

For the effect of wastewater, we didn't distinguish the influence from the Pearl River and Hong Kong local pollution in our study as we aimed to investigate the effect of the long-term variations

in physical conditions and nutrient levels within the region from a general perspective. As for the role of the Pearl River, there were indeed differences among stations, which were also shown by different correlations of DO with other external variables in each station. It might be interesting to dig into this issue, but our present work mainly focused on the whole status of low-oxygen conditions in the selected region, rather than dealing with the spatial differences.

**Reference**

Cui, Y., Wu, J. X., Ren, J., and Xu, J.: Physical dynamics structures and oxygen budget of summer hypoxia in the Pearl River Estuary: Dynamical structures and oxygen budget of hypoxia, Limnology and Oceanography, 64, 10.1002/lno.11025, 2018.

Hu, J., Zhang, Z., Wang, B., and Huang, J.: Long-term spatiotemporal variations and expansion of low-oxygen conditions in the Pearl River estuary: A study synthesizing observations during 1976–2017, 10.5194/bg-2020-480, 2021.

Li, D., Gan, J., Hui, C., Yu, L., Liu, Z., Lu, Z., Kao, S.-j., and Dai, M.: Spatiotemporal Development and Dissipation of Hypoxia Induced by Variable Wind-Driven Shelf Circulation off the Pearl River Estuary: Observational and Modeling Studies, Journal of Geophysical Research: Oceans, 126, e2020JC016700, https://doi.org/10.1029/2020JC016700, 2021.

Li, X., Lu, C., Zhang, Y., Zhao, H., Wang, J., Liu, H., and Yin, K.: Low dissolved oxygen in the Pearl River estuary in summer: Long-term spatio-temporal patterns, trends, and regulating factors, Marine Pollution Bulletin, 151, 110814, https://doi.org/10.1016/j.marpolbul.2019.110814, 2020.

Lu, Z., Gan, J., Dai, M., Liu, H., and Zhao, X.: Joint effects of extrinsic biophysical fluxes and intrinsic hydrodynamics on the formation of hypoxia west off the Pearl River Estuary, Journal of Geophysical Research: Oceans, 123, 6241-6259, 2018.

Lui, H.-K., Chen, C.-T. A., Hou, W.-P., Liau, J.-M., Chou, W.-C., Wang, Y.-L., Wu, C.-R., Lee, J., Hsin, Y.-C., and Choi, Y.-Y.: Intrusion of Kuroshio Helps to Diminish Coastal Hypoxia in the Coast of Northern South China Sea, Frontiers in Marine Science, 7, 565952, 10.3389/fmars.2020.565952, 2020.

Qian, W., Gan, J., Liu, J., He, B., Lu, Z., Guo, X., Wang, D., Guo, L., Huang, T., and Dai, M.: Current status of emerging hypoxia in a eutrophic estuary: The lower reach of the Pearl River Estuary, China, Estuarine, Coastal and Shelf Science, 205, 58-67, https://doi.org/10.1016/j.ecss.2018.03.004, 2018.

Shi, Z., Liu, K., Zhang, S., Xu, H., and Liu, H.: Spatial distributions of mesozooplankton biomass, community composition and grazing impact in association with hypoxia in the Pearl River Estuary, Estuarine, Coastal and Shelf Science, 225, 106237, https://doi.org/10.1016/j.ecss.2019.05.019, 2019.

Wang, B., Hu, J., Li, S., and Liu, D.: A numerical analysis of biogeochemical controls with physical modulation on hypoxia during summer in the Pearl River Estuary, 1-31 pp., 10.5194/bg-2016-454, 2016.

Wang, B., Hu, J., Li, S., Yu, L., and Huang, J.: Impacts of anthropogenic inputs on hypoxia and oxygen dynamics in the Pearl River estuary, Biogeosciences, 15, 6105-6125, 10.5194/bg-15-6105-2018, 2018.

Yin, K., Lin, Z., and Ke, Z.: Temporal and spatial distribution of dissolved oxygen in the Pearl River Estuary and adjacent coastal waters, Continental Shelf Research, 24, 1935-1948, https://doi.org/10.1016/j.csr.2004.06.017, 2004.

---

## Author Response (AR1)

We wish to thank the editor and two referees for the constructive comments and suggestions which are helpful to the revision of our manuscript. Detailed responses to all comments are given below (responses are shown in blue and relevant changes are marked in red in the revised manuscript).

**Editorial comments:**

Your plan for revising the manuscript in response to the reviewers' appears to address the major issues. However, there are several aspects of your planned revisions that are of especially high importance and you need to be sure you thoroughly address these. First, it is not clear in some of your responses to comments what actually will be done to the manuscript text versus you discussing the comment as if you were talking to me. Your responses were detailed but what is important is that the manuscript gets revised according to the your responses. For example, discussing the comment in 2 paragraphs in your plan should likely led to more changes to the manuscript then the addition of a few sentences. You have some important information in the responses that should be included in the revised manuscript. Second, both reviewers questioned quite vigorously whether the monitoring data you selected to analyze was representative of the estuary. You need to make a very convincing case in the manuscript that your analysis reflects the broader geographic area. Third, and related to the second issue, is that you should bring into the manuscript a succinct synthesis of other studies that examined similar data and/or similar questions and how they compare to your analysis. Fourth, a reviewer wanted more details on the interpolation used and clearer description of how the PCA was done and the interpretation of the PCA results. Finally, adding the regression results is appropriate but the coefficients alone do not tell you very much; consider reporting effect sizes. For more information on this, refer to Smith, E.P., 2020. Ending reliance on statistical significance will improve environmental inference and communication. Estuaries and Coasts, 43(1), pp.1-6. I look forward to seeing the revised manuscript.

Response: As suggested, we have revised our manuscript and added more details of the revisions in our response as follows.

(1) For the representative of Hong Kong data in use, we would like to emphasize our consideration and make adjustment in the revised manuscript accordingly. Based on in-situ observations in the PRE, low-oxygen conditions have been frequently observed in the coastal waters off Hong Kong since the 1980s (Li et al., 2021; Li et al., 2020; Qian et al., 2018; Lu et al., 2018) and this region was close to a hotspot area of low-oxygen conditions in the eastern PRE (Hu et al., 2021). After consideration, we have changed the phase "the eastern PRE" into "the coastal waters off Hong Kong" in the revised manuscript for more precise description, given the fact that the data are indeed inappropriate to represent the whole eastern PRE, which needs more data in boarder coverage. Nevertheless, it is important to note that the quantitative investigation of deoxygenation conducted for the Hong Kong waters in our study could be a significant supplement to the

understanding of the low-oxygen conditions in the PRE. Therefore, we have added more description of the merits of Hong Kong data and water quality conditions in Hong Kong to clarify our research background in the Introduction part (please see Page 3, Line 76-90 in the revised manuscript).

(2) For the description of our research methods, we have added the description about the interpolation and estimation of low-oxygen areas and thicknesses in the Materials and Methods part (please see Page 4, Line 111-120 in the revised manuscript). Besides, we have provided more detailed results of the PCA analysis in Page 5, Line 126-131 and Appendix of the revised manuscript.

(3) For the regression results, we have provided more details of different test results including sample sizes and their effect sizes (e.g. $R^2$ and Pearson correlation coefficient) in Page 5, Line 152-158 and Appendix of the revised manuscript. In brief, although the fitting effects of training dataset were similar, their predictive effects of testing dataset varied. We have also added a flow-chart and enhanced the description to make the sampling processes clearer based on the reviewer's suggestions.

(4) For the combination of previous studies, we have added more discussion on the effects of wind, freshwater discharge, and nutrient loads on deoxygenation in combination with the relevant findings of the previous studies. Specifically, we compared the effect of wind speed and direction among our results and previous studies, showing that the wind direction played an important role in short-term low-oxygen events, but contributed less from the interannual and long-term perspective. Besides, we have made more comparison about the effect of nutrients and freshwater discharge among the Hong Kong waters and other subregions of the PRE as well as other coastal systems (e.g., Chesapeake Bay, the northern Gulf of Mexico). Please see details in Page 9-10 in the revised manuscript.

(5) In addition, we have revised several graphs (such as Figures 1, 4, and 5 in the revised manuscript) and their figure captions to make the information clearer.

Again, thank you very much for providing these comments and suggestions.

**Anonymous Referee #1**

**General Comments**

Chen et al. evaluated long-term patterns in DO in the eastern Pearl River Estuary (PRE) across seasons and regions, computed an aggregated metric of low DO, and then tested possible controlling factors of it with multiple regression. They found dissolved nitrogen and wind speed were the most explanatory variables for interannual variations and long-term trends. They use additional water quality observations to evaluate the changes to the system over time and hypothesize shifts in the system dynamics. Overall, this is a very interesting study making good use of a long-term data set to evaluate long-term change. I appreciate their thorough treatment of the data both spatially and temporally. My major comments involve clarifying the methods and what is represented in some of the graphics. Clarification is needed throughout as to which months of data are included in different average results and how the data is aggregated to represent "summer". In addition, more clarification is needed on the PCA approach as well as some re-organization of which information is presented in the Methods or Results.

Response: We are grateful to the reviewer for the positive comments and recognition of our work. As suggested, we have provided clearer description to clarify the methods and results as follows.

(1) Firstly, we have enhanced the description about the selection of Hong Kong long-term dataset in the Introduction part of the revised manuscript. In brief, we have described the merits of the Hong Kong data and water quality conditions in the coastal waters off Hong Kong in order to make the research background clearer. Pease see details in our response to Comment #1 below.

(2) In the Materials and Methods part, we have added the description of interpolation and the methods to estimate low-oxygen areas and thicknesses. Please see details in our response to Comment #6 below. Besides, we have provided more description of input variables and detailed results of the PCA analysis, including variance explained and the feature matrix. Please see details in our response to Comment #3 below.

(3) In the Result part, we have enhanced the description and discussion of the different test results in multiple regressions. Of all the established models, we found similar fitting effect of training datasets (e.g. $R^2_{train}$) and coefficients of the four variables, but the predictive skills in testing dataset (e.g. $R^2_{test}$) varied in a large range. We have also added a flow-chart and enhanced the description to make the sampling processes clearer. Please see details in our response to Comment #5 below.

(4) We have revised some graphs and figure captions to make the information clearer, such as the clarification of "summer". Please see details in our response to Comments #2, 6, 7, 8, 9, and 10 below.

**Specific Comments:**

1. **Lines 60 – 81:** Within this section, please incorporate the reasoning for your focus on the Eastern PRE. Can you describe whether this region was selected from the larger PRE because this is where the longest term data is, or is it because this is where the lowest oxygen occurs? It would provide more context if you included some description in the Introduction about how water quality in this eastern region compares to the rest of the estuary.

Response: In fact, the merits on the length and coverage of Hong Kong dataset were an important reason. In addition, low-oxygen and hypoxic events have frequently occurred in the coastal waters off Hong Kong, which was another more important consideration. As Hu et al. (2021) pointed out, observational data on DO and other water quality parameters in the PRE were generally scarce in time and space, while the sampling time periods and sites and sometimes the water quality measurement methods were quite different between available datasets. These limitations and uncertainties inherent in the historical data (including the observations during 1976-2017 compiled by Hu et al., 2021) brought out great obstacles for quantifying the long-term deoxygenation trend in the PRE. By comparison, the monthly data used in our present study, collected in the coastal waters off Hong Kong, have significant merits in terms of temporal coverage (over three decades) and consistency of sampling locations. Therefore, these data with good spatiotemporal continuity enabled us to better estimate the long-term and interannual variations in low-oxygen conditions without concerns on the uncertainties that would arise from the usage of different data sources. Besides, various oxygen-related parameters (e.g., chlorophyll-*a* concentrations) were measured as well and could be used to discern the key factors controlling the long-term oxygen changes. More importantly, low-oxygen and hypoxic conditions have been reported in the coastal waters adjacent to Hong Kong (Yin et al., 2004; Su et al., 2017; Shi et al., 2019) since the 1980s, and this region was close to a hotspot area of low-oxygen conditions in the eastern PRE (Figure r1). Previous studies have also showed that the dominant deoxygenation mechanisms varied among subregions in the PRE. While Modaomen Bay and the upper region of Lingdingyang Bay were more heavily influenced by terrestrial pollutants (Li et al., 2020; Wang et al., 2017; Wang et al., 2018), low-oxygen conditions observed in the coastal waters off Hong Kong were regulated by the joint effect of physical process (Li et al., 2021) and eutrophication (Qian et al., 2018). Therefore, the study of deoxygenation in the Hong Kong waters could be a significant supplement to the understanding of the low-oxygen conditions in the whole PRE.

[Figure]

Figure r1 Observation sites during 1976-2014 in the PRE in previous studies (Hu et al., 2021) and station sites in our work (a). Note that black dots represent observation sites in previous studies and blue, red, green dots represent station sites in the northwestern, southern, and eastern subregions of Hong Kong waters, respectively. The incidence of low-oxygen (b1) and oxygen-deficiency (b2) conditions at the bottom of the PRE in summer during 1976-2014 (Hu et al., 2021).

Based on the suggestion from the reviewers and editor, we agreed to change the phase of "the eastern PRE" to "the coastal waters off Hong Kong" in the revised manuscript for more precise description. We have provided more descriptions for selection of the dataset and water quality conditions of Hong Kong waters in Page 3, Line 76-90 of the revised manuscript as follows:

*"In this study, we utilize observational oxygen and related data collected by the Hong Kong Environmental Protection Department (HKEPD) at certain coastal sites off Hong Kong (see details in section 2.1 below), which have significant merits in terms of temporal coverage (~30 years) and consistency of sampling locations, to perform a quantitative analysis on the long-term oxygen changes (trend and interannual variability) in the region. Moreover, we also aim to discern the key factors controlling the interannual variability and long-term trends in the low-oxygen conditions and to quantify the relative contribution of each primary factor using multiple regression models (Murphy et al., 2011; Forrest et al., 2011; Wang et al., 2021). It is important to note that the HKEPD data with good spatiotemporal continuity allowed us to better estimate the long-term deoxygenation in the coastal waters off Hong Kong, which was close to a hotspot area of low-oxygen conditions in the eastern PRE (Hu et al., 2021) and subject to frequent occurrences of low-oxygen and hypoxic events as well (Yin et al., 2004; Su et al., 2017; Shi et al., 2019). In addition, previous studies have showed that the dominant deoxygenation mechanisms varied between subregions in the PRE; for instance, the low-oxygen conditions in Modaomen Bay were primarily determined by terrestrial pollutant inputs (Li et al., 2020; Wang et al., 2017; Wang et al., 2018), whereas those in the coastal waters off Hong Kong were largely controlled by the joint effect of physical processes (e.g. convergence of water masses (Li et al., 2021)) and eutrophication (Qian et al., 2018). Therefore,*

*the extensive investigation on deoxygenation performed here for the Hong Kong waters is a significant supplement to the understanding of low-oxygen conditions for the whole PRE."*

2. **Line 104:** Please describe what spatial interpolation approach was used in MATLAB for the interpolations. Also, since you have land in between some of the stations, how did the method deal with that? It would be helpful to show what the region looks like in vertical cross-section as a 2nd panel of Figure 1 with the sample locations and depths indicated with dots. This would be like one of the panels of Figure A2, showing which depths each station is sampled at. This would be a helpful way to visualize the depths at each station.

Response: We estimated the vertical distribution of each water quality variable by using the "Natural-Neighbor" method, which has been widely applied in geophysics studies for its high spatial autocorrelation, in MATLAB. Regarding the island between stations NM8 and SM20, we first performed the spatial interpolations directly with all the observed data and then masked out the area cover by the island, which was roughly estimated based on its size (please note that the topographic data of the island was not available). Such a treatment has little influence on the estimation of vertical low-oxygen areas because low-oxygen conditions were seldom found in stations NM8 and SM20. Moreover, same treatment procedure was applied in each month of 25 years to make it consistent when investigating the interannual variations in low-oxygen conditions. We have enhanced the description of interpolation and the treatment of the island in Page 4, Line 112-121 as follows:

*"Firstly, the observed DO profiles were interpolated by "Natural-Neighbor" method through MATLAB along the three subregions with a grid resolution of 600 m (distance) × 0.3 m (depth). The total areas of DO below 4 mg/L, 3 mg/L, and 2 mg/L were then calculated as the cross-sectional areas of low oxygen, oxygen deficiency, and hypoxia, respectively. The associated layer thickness was defined as the averaged thickness of the grids with DO below the corresponding levels (i.e. 4 mg/L, 3 mg/L, and 2 mg/L). Regarding the island between stations NM8 and SM20, the spatial interpolations were performed directly with all the observed data and then the areas covered by the island were masked out roughly based on its size (Figures 2, 3, A4), as the topographic data of the island was not available. Such a treatment has little influence on the estimation of vertical low-oxygen areas because low-oxygen conditions were seldom found in stations NM8 and SM20. Moreover, same treatment procedure was applied in each month of 25 years to make it consistent when investigating the interannual variations in low-oxygen conditions."*

Moreover, as suggested, we have updated Figure 1 by adding subgraphs to show the location and depth of each station (please see Figure r2 below).

[Figure]

Figure r2. (a) Map of the Pearl River Estuary (PRE) and monitoring stations in the coastal waters off Hong Kong. Note that the blue, red, green dots represent stations in the northwestern, southern, eastern subregions of Hong Kong, respectively. The red triangle denotes the location of Waglan Island automatic weather station and the purple dots indicate the location of cities in the Guangdong-Hong Kong-Macao Greater Bay Area. (b) Four subgraphs showing the vertical distributions of mean DO concentrations in winter (Dec, Jan, Feb), spring (Mar, Apr, May), summer (Jun, Jul, Aug) and autumn (Sep, Oct, Nov) during 1994-2018.

3. **Lines 108-120:** This discussion of the PCA needs modification. Please include a table of the variables used in the PCA. I kept having to look back in the text to see how "low oxygen", "Area3," etc, were defined. I'd suggest including just that table and a description of the approach here in the Methods section. The resulting equation (Line 117) and description of it should probably be moved to the Results section. Also, please summarize the rest of the PCA results (in an appendix table), such as what % of variance the other components had, and what their weights were.

Response: As suggested, we have added tables of the input variables used and detailed results of the PCA analysis (please see Tables r1 and r2 below), and have moved Equation (1) and its associated description to the Results section (section 3.2) in the revised manuscript. In addition, we have provided more details on the rest of the PCA results in Page 5, Line 126-131 and Appendix of the revised manuscript as follows:

*"The results of PCA analysis (Table A2) showed that the first component explained most of variance (86.40%) for the six input variables, while the remaining components explained less variance (13.60%). The first component was highly correlated with the interannual variations of the cross-sectional areas (with a correlation coefficient r of 0.96, p < 0.01) and the thickness (r = 0.96, p < 0.01) of low oxygen as well as the bottom DO concentrations (r = -0.90, p < 0.01), and it was thereafter referred as Low-oxygen Index (LOI, Equ. 1) to describe the interannual severity of low-oxygen conditions comprehensively."*

Table r1. Description of variables in the Principal Component Analysis (PCA)

| Variables in PCA | Description |
|---|---|
| $DO_{mean}$ | Spatial average value of DO concentrations in bottom of each year during 1994-2018 |
| $DO_{min}$ | Spatial minimum value of DO concentrations in bottom of each year during 1994-2018 |
| $Area_4$ | Cross-sectional area of low-oxygen (DO<4 mg/L) of each year during 1994-2018 |
| $Area_3$ | Cross-sectional area of oxygen-deficiency (DO<3 mg/L) of each year during 1994-2018 |
| $Thickness_4$ | Cross-sectional thickness of low-oxygen of each year during 1994-2018 |
| $Thickness_3$ | Cross-sectional thickness of oxygen-deficiency of each year during 1994-2018 |
| Low-oxygen Index (LOI) | First principal component of PCA dimension (86.40% of variation) for measuring interannual variations in scope and intensity of oxygen conditions |

Table r2. Total variance explained and the feature matrix of the first component in the PCA

| Component | Eigenvalues | Variance explained (%) | Accumulative variance explained (%) | Feature matrix | Proportion |
|---|---|---|---|---|---|
| 1 | 5.184 | 86.404 | 86.404 | $DO_{mean}$ | -0.903 |
| 2 | 0.382 | 6.370 | 92.774 | $DO_{min}$ | -0.880 |
| 3 | 0.285 | 4.745 | 97.519 | $Area_4$ | 0.960 |
| 4 | 0.149 | 2.481 | 100.000 | $Area_3$ | 0.935 |
| 5 | $-1.079\times10^{-16}$ | $-1.798\times10^{-15}$ | 100.000 | $Thickness_4$ | 0.960 |
| 6 | $-4.472\times10^{-16}$ | $-7.454\times10^{-15}$ | 100.000 | $Thickness_3$ | 0.935 |

4. **Line 125:** show an equation to describe this standardization

Response: As suggested, we have added an equation to describe this standardization in Page 5, Line 149-151 of the revised manuscript as follows:

"

$$Cst_i = C_i \times \frac{SD_i}{SD_{LOI}}$$ (2)

*Where $Cst_i$ and $C_i$ represent the regression coefficients of WS, flow, T and DIN after and before standardization, respectively; $SD_i$ represent the standard deviation of WS, flow, T and DIN; $SD_{LOI}$ represents the standard deviation of LOI.*"

5. **Lines 123-134:** There need to be some discussion of these different test results in the Results section.

Response: As suggested, we have provided more details on different test results of regressions

(please see Figure r3 and Table r3 below). It can be seen from Figure r3 that the ensemble means of fitting Low-oxygen Index (LOI) for different combinations of training and testing datasets were similar, but the variance explained by the fitting and the regression coefficients were different (Table r3). For the regression models with lower coefficients of determination ($R^2$) (Figure r3c), the fitting LOI both in training dataset and testing dataset varied in a relatively larger range. To provide a more robust data fitting result, we chose the cases with excellent performances (indicated by $R^2$ over 0.6 both in the training and testing datasets) for formal analysis. Based on the reviewer's suggestion, we have added the description on sampling for fitting models in Page 5, Line 152-158 of the revised manuscript as follows, and have provided Figure r3 and Table r3 in Appendix.

*"In addition, the dataset was randomly split into a training dataset (70%) and a testing dataset (30%) in order to provide a more robust data fitting with estimates on the uncertainties arising from different data selections. Consequently, over 480,700 combinations of training and testing datasets were generated randomly from this splitting process and were used to build up a variety of regression models. Coefficient of determination ($R^2$) was used to measure the fitting effect in training and testing datasets. Of all the established models, the fitting effect of training datasets (e.g. $R^2_{train}$) and coefficients of the four variables were similar, but the predictive skills in testing dataset (e.g. $R^2_{test}$) varied in a large range (Figure A5, Table A4). Besides, larger standard deviation occurred in coefficients in cases with worse testing effects."*

[Figure]

Figure r3. Combined fitting results of the regression models with different combinations of training and testing datasets. $R^2$ in (a) were greater than or equal to 0.6 both in training and testing datasets.

$R^2$ in (b) were less than 0.6 both in training and testing datasets. Fitting results of total samples were in (c). Note that the red hollow dots denote the LOI estimated based on observational data, while the blue solid dots and the gray patch represent the mean values and ranges of the fitted LOI in the selected regression cases, respectively.

Table r3. Regression coefficients of wind speed (WS), freshwater discharge (flow), bottom temperature (T) and surface DIN (DIN) of different sample datasets (Mean±Std); $R^2$ and Pearson correlation coefficient (r) of training and testing dataset in different sample datasets (Mean±Std)

| Fitting cases | Sample size | Coefficient of WS | Coefficient of flow | Coefficient of T | Coefficient of DIN |
|---|---|---|---|---|---|
| $R^2_{train} \geq 0.6$ & $R^2_{test} \geq 0.6$ | 56010 | -0.39±0.12 | -0.14±0.12 | -0.11±0.08 | 0.49±0.12 |
| $R^2_{train} < 0.6$ or $R^2_{test} < 0.6$ | 424690 | -0.37±0.14 | -0.17±0.17 | -0.12±0.11 | 0.44±0.15 |
| Total samples | 480700 | -0.37±0.14 | -0.16±0.16 | -0.12±0.11 | 0.45±0.14 |
| | p value | $R^2_{train}$ | $R^2_{test}$ | $r_{train}$ | $r_{test}$ |
| $R^2_{train} \geq 0.6$ & $R^2_{test} \geq 0.6$ | 0.008±0.003 | 0.64±0.08 | 0.70±0.08 | 0.80±0.02 | 0.83±0.05 |
| $R^2_{train} < 0.6$ or $R^2_{test} < 0.6$ | 0.012±0.014 | 0.64±0.24 | 0.45±0.24 | 0.80±0.05 | 0.63±0.24 |
| Total samples | 0.011±0.013 | 0.64±0.24 | 0.48±0.24 | 0.80±0.05 | 0.65±0.24 |

6. **Figure 4a:** can you describe the values plotted here more? Is the minimum, mean and range just from the bottom observations, or is it generated from the interpolation?

Figure 4 (b) and (c)– We need information on the spatial interpolation to get the area and thickness. Also, if samples are collected every month, it is unclear what the bars in (b) and (c) represent. Are they the average of each month's spatially-aggregated values? If so, please put range bars on each bar to show the range across the summer months. Or pick one month to show.

Response: The mean and minimum DO values shown in Figure 4a are the spatiotemporally average and minimum DO concentrations for the bottom observations (using all the data from 10 stations in June, July, and August, i.e. 30 data points in total for each year). Also, the grey patch showed the minimum-to-maximum range of the observations in each year during 1994-2018. We have clarified this in the caption of Figure 4 in the revised manuscript.

With respect to the areas and thickness of oxygen conditions, they were estimated by the methods mentioned in section 2.2. Specifically, we calculated the areas and thicknesses in June, July, and August of each year, respectively, and then computed the corresponding summer means

(results shown in Figure 4b-c) by averaging the areas and thicknesses of the three months. As suggested, we have clarified these estimations and revised Figure 4b-c by adding error bars to show the range across the summer months (please see the revised figure below):

[Figure]

Figure r4. Interannual variations in the spatiotemporally (10 stations in June, July and August) mean and minimum concentrations of observed DO at the bottom (a), the cross-sectional areas (b) and layer thicknesses (c) of low-oxygen conditions, and LOI (e) in summer during 1994-2018. Note that the coloured bars in (b) and (c) show the mean values of three summer months, while the black thin error bars represent the range across three summer months.

7. **Figure A2:** Similarly to Figure 4, specify which month of the summer these plots are for. If they are average of all the summer cruises, please justify that approach.

Response: Please note that this figure has been changed to Figure A4 in the revised manuscript. As for the vertical DO distributions in this figure, they are averages of all the summer cruises. Specifically, we first averaged the observational data in the three summer months to obtain the summer means at the surface, middle, and bottom, respectively, and then interpolated them onto the profile grids to get the summertime vertical DO distributions. We have clarified this in the caption of Figure A4 in the revised manuscript.

8. **Figure 5:** The min and mean DO symbols in legend seem switched.

Response: We have corrected this mistake.

9. **Figure 5:** I'd like to see the surface and bottom graphs with the same vertical scale (0 to 10).It can be confusing to have them different when they are right next to each other.

Response: As suggested, we have used the same vertical scale (0 to 10) in the surface and bottom graphs as follows.

[Figure]

Figure r5. Long-term trends of the mean and minimum values of observed DO at the surface and bottom waters in three summer months for all the stations (a-b) and for the northwestern (c-d), southern (e-f), and eastern (g-h) subregions, and long-term trends of the cross-sectional areas of low oxygen (i) and oxygen deficiency (j).

10. **Figure 5:** I'm unsure from the descriptions as to how the mean and minimum were calculated with multiple stations and months of the summer. Is the minimum the absolute minimum

observed in that region in the summer, or an average of the lowest value across the stations? Also is the mean a spatial and temporal mean across the summer?

Response: As mentioned earlier (please see our response to Comment #6), the mean and minimum DO values shown in Figure 5 are the spatiotemporally average and minimum DO concentrations for the bottom observations (using all the data from 10 stations in June, July, and August, i.e. 30 data points in total for each year). Indeed, the minimum is the absolute DO minimum observed in the region in three summer months, and the mean is the spatiotemporal mean across the summer. We have revised the caption of Figure 5 to make it clear as follows:

"*Figure 5. Long-term trends of the mean and minimum values of observed DO at the surface and bottom waters in three summer months for all the stations (a-b) and for the northwestern (c-d), southern (e-f), and eastern (g-h) subregions, and long-term trends of the cross-sectional areas of low oxygen (i) and oxygen deficiency (j).*"

11. **Figure 6:** The really high values in the range in recent years in July are worth mentioning. Is that just one location that is causing that range to increase, or is it some indication of increased variability?

Response: For the larger range of DO in July after 2012, it exhibited increased spatial variability. As shown in Figure r6 below, in July after 2012, high DO was constantly observed at the bottom of stations NM6, NM8, and SM20, while low DO was observed at the bottom of SM18 and SM19. We could find some explanations from the vertical distributions of Chl *a* concentrations (please see Figure r6 below). The high Chl *a* in NM6, NM8, and SM20 revealed that phytoplankton flourished within the region. Due to the shallow depth of these stations, high DO resulted from photosynthesis was observed at the surface and the bottom. However, stations located in the downstream (e.g. SM18 and SM19) would receive much organic matter transported by hydrodynamic processes. Driven by stronger water-column stratification therein, low-oxygen conditions often occurred at bottom in SM stations. After 2012, high Chl *a* occurred in July with more frequency and larger intensity, which could be a main reason for larger range of DO at the bottom.

[Figure]

Figure r6. vertical distributions of DO and Chl *a* in July and August during 2012-2018.

12. **Line 233:** A diagram or flow-chart that describes the sampling and cases used in the regression analysis to get to the results would help my understanding (and probably other readers) of the methods. This could go in the Appendix.

Response: As suggested, we have added a flowchart to describe the fitting and sampling processes in Appendix of the revised manuscript as follows:

[Figure]

Figure r7. Flowchart describing the fitting of LOI and the cases sampling used for analysis.

13. **Figure 9:** The wind speed decrease over time seems very large. Because the results indicate this is an important variable, this deserves more discussion or investigation. If the authors already know other work that has investigated decreasing wind speeds, please cite it and describe briefly. But if there is no other research explaining this wind decrease, it would be a good idea to double-check the data and be sure that it is not an artifact of sampling dates or density shifting or sensor height changing.

Response: Actually, we could find supports on the decrease of wind speed both in the Pearl River Basin (Zhang et al., 2019) and the northern South China Sea (Gao et al., 2020). Previous studies suggested that due to the weakening of atmospheric cycle activities and the East Asian monsoon (Xu et al., 2006; Zhang et al., 2009), the annual average wind speed in the Pearl River Basin exhibited a significant declining trend (at a rate of -0.003 m/s per year) during 1960-2016 (Zhang et al., 2019). In addition, previous studies showed that the long-term increase of air temperature in the Pearl River Delta region contributed to air stability and weakened the intensity of tropical cyclone (Chen et al., 2020), which would lead to a decline of summertime wind speed in the PRE. In the northern South China sea, the summer average wind speed decreased at a rate of 0.05 m/s per year during 2004-2020 (Gao et al., 2020), which was close to that in the eastern PRE (-0.03 m/s per year).

14. **Appendix Figure A1:** is important b/c it doesn't suffer from any possible aggregation or averaging bias. It might be useful to make an addition panel that shows how the bottom summer counts have changed over time – maybe make one for the first half and one for the 2nd half of the record. This could also show if there's a spatial shift.

Response: Please note that this figure has been changed to Figure A3 in the revised manuscript. Based on the reviewer's suggestion, we have investigated changes in the occurrence of bottom low-oxygen conditions in summer during two different periods (please see Figure r8 below) to examine if there is a spatial shift. As shown, the low-oxygen and hypoxic conditions were more severe during 2006-2018 when compared to those during 1994-2005, implying a deoxygenation pattern over time. However, there is no spatial shift found between the two periods. Stations NM5 and SM18 had the highest intensity of low-oxygen events in both periods.

[Figure]

Figure r8. Frequencies of occurrence of low-oxygen and hypoxic events in four seasons at the bottom (1994-2018, 1994-2005, 2006-2018) layer.

**Technical Corrections:**

(1) **Abstract, Line 15** – change "was" to "were"

Response: We have corrected it as suggested.

(2) **Abstract, line 17** – suggest changing "through the principal component analysis" to something else. Maybe "as a result of a principal component analysis"

Response: We have corrected it as suggested.

(3) **Abstract, line 25** – It is unclear what "It" refers to in this sentence. Please re-write.

Response: We have revised it as follows:

*"The deteriorating eutrophication has driven a shift in the dominant source of organic matter from terrestrial inputs to in situ primary production, which has probably led to an earlier onset of*

*hypoxia in summer."*

(4) **Abstract, last sentence** – the phrase "in the context of" is fairly awkward. Consider re-wording this sentence to make your summary stronger.

Response: We have revised it as follows:

*"In summary, the Hong Kong waters has undergone considerable deterioration of low-oxygen conditions driven by substantial changes in anthropogenic eutrophication and external physical factors."*

(5) **Intro, Line 33** – suggestion you use "organisms" instead of "creature"

Response: We have corrected it as suggested.

(6) **Intro, Line 43-45** – Simplify (or remove) this sentence since the next few sentences cover a lot about oxygen depletion. I'd suggest just "Terrestrial organic matter discharged to estuaries can lead to intense microbial respiration."

Response: We have corrected it as suggested.

(7) **Intro, Line 54:** For the Ni et al. 2020 paper, it is important to change "ocean" to "estuary." They did not study the external impact of the Atlantic Ocean warming on the Chesapeake Bay.

Response: We have corrected it as suggested.

(8) **Methods, Lines 84-93** – Who collected this data?

Response: The data were provided by the Environmental Protection Department of Hong Kong. We have made it clear in the Materials and Methods section.

(9) **Results, Line 144** – I do not think the word "varied" is correct here.

Response: We have revised the sentence to *"The summertime temperature fluctuated between 28.21 ±1.19 °C..."* . All the relevant phrases in the manuscript have been revised accordingly.

(10) **Results, Line 168** – wording like this sentence can be simplified. You could just start with "DO concentrations exhibited significant…"

Response: We have corrected it as suggested.

(11) **Results, Line 192, and other places** – The phrase "DO content" is not something I've seen very much before in the hypoxia literature. I'd suggest using "DO concentrations" or just "DO".

Response: We have corrected it as suggested.

(12) **Figure 6** – It would be helpful to use the same open circles for the blue symbols as in Figure 5.

Response: As suggested, we have changed the legends of $DO_{mean}$ and $DO_{min}$ in Figure 6 and made them consistent with those in Figure 5.

(13) **Discussion, Lines 297-298:** Please revise the sentence that starts with "As quantified by statistic methods…" to work on the wording. Maybe "Our analysis showed that increasing DIN…"

Response: We have corrected it as suggested.

Reference

Chen, J., Wang, Z., Tam, C.-Y., Lau, N.-C., Lau, D.-S. D., and Mok, H.-Y.: Impacts of climate change on tropical cyclones and induced storm surges in the Pearl River Delta region using pseudo-global-warming method, Scientific Reports, 10, 1965, 10.1038/s41598-020-58824-8, 2020.

Gao, N., Ma, Y., Zhao, M., Zhang, L., Zhan, H., and He, Q.: Quantile Analysis of Long-Term Trends of Near-Surface Chlorophyll-a in the Pearl River Plume, Water, 12, 1662, 10.3390/w12061662, 2020.

Hu, J., Zhang, Z., Wang, B., and Huang, J.: Long-term spatiotemporal variations and expansion of low-oxygen conditions in the Pearl River estuary: A study synthesizing observations during 1976–2017, 10.5194/bg-2020-480, 2021.

Li, D., Gan, J., Hui, C., Yu, L., Liu, Z., Lu, Z., Kao, S.-j., and Dai, M.: Spatiotemporal Development and Dissipation of Hypoxia Induced by Variable Wind-Driven Shelf Circulation off the Pearl River Estuary: Observational and Modeling Studies, Journal of Geophysical Research: Oceans, 126, e2020JC016700, https://doi.org/10.1029/2020JC016700, 2021.

Li, X., Lu, C., Zhang, Y., Zhao, H., Wang, J., Liu, H., and Yin, K.: Low dissolved oxygen in the Pearl River estuary in summer: Long-term spatiotemporal patterns, trends, and regulating factors, Marine Pollution Bulletin, 151, 110814, https://doi.org/10.1016/j.marpolbul.2019.110814, 2020.

Lu, Z., Gan, J., Dai, M., Liu, H., and Zhao, X.: Joint effects of extrinsic biophysical fluxes and intrinsic hydrodynamics on the formation of hypoxia west off the Pearl River Estuary, Journal of Geophysical Research: Oceans, 123, 6241-6259, 2018.

Qian, W., Gan, J., Liu, J., He, B., Lu, Z., Guo, X., Wang, D., Guo, L., Huang, T., and Dai, M.: Current status of emerging hypoxia in a eutrophic estuary: The lower reach of the Pearl River Estuary, China, Estuarine, Coastal and Shelf Science, 205, 58-67, https://doi.org/10.1016/j.ecss.2018.03.004, 2018.

Shi, Z., Liu, K., Zhang, S., Xu, H., and Liu, H.: Spatial distributions of mesozooplankton biomass, community composition and grazing impact in association with hypoxia in the Pearl River Estuary,

Estuarine, Coastal and Shelf Science, 225, 106237, https://doi.org/10.1016/j.ecss.2019.05.019, 2019.

Su, J., Dai, M., He, B., Lifang, W., Gan, J., Guo, X., Zhao, H., and yu, F.: Tracing the origin of the oxygen-consuming organic matter in the hypoxic zone in a large eutrophic estuary: The lower reach of the Pearl River Estuary, China, Biogeosciences, 14, 4085-4099, 10.5194/bg-14-4085-2017, 2017.

Wang, B., Hu, J., Li, S., and Liu, D.: A numerical analysis of biogeochemical controls with physical modulation on hypoxia during summer in the Pearl River estuary, Biogeosciences, 14, 2979-2999, 10.5194/bg-14-2979-2017, 2017.

Wang, B., Hu, J., Li, S., Yu, L., and Huang, J.: Impacts of anthropogenic inputs on hypoxia and oxygen dynamics in the Pearl River estuary, Biogeosciences, 15, 6105-6125, 10.5194/bg-15-6105-2018, 2018.

Xu, C.-Y., Gong, L., Tong, J., Chen, D., and Singh, V.: Analysis of spatial distribution and temporal trend of reference ET in Changjiang catchments, Journal of Hydrology, 327, 81-93, 10.1016/j.jhydrol.2005.11.029, 2006.

Yin, K., Lin, Z., and Ke, Z.: Temporal and spatial distribution of dissolved oxygen in the Pearl River Estuary and adjacent coastal waters, Continental Shelf Research, 24, 1935-1948, https://doi.org/10.1016/j.csr.2004.06.017, 2004.

Zhang, T., Chen, Y., and U, K.: Quantifying the impact of climate variables on reference evapotranspiration in Pearl River Basin, China, Hydrological Sciences Journal, 64, 1-13, 10.1080/02626667.2019.1662021, 2019.

Zhang, X., Ren, Y., Yin, J., Lin, Z., and Zheng, D.: Spatial and temporal variation patterns of reference evapotranspiration across the Qinghai-Tibetan Plateau during 1971–2004, J. Geophys. Res, 114, 10.1029/2009JD011753, 2009.

**Anonymous Referee #2**

**General Comments**

1. The manuscript by Chen et al. provides data analysis (mostly statistical) of the water quality observations in the southeastern part of the Pearl River Estuary (PRE). Several recent studies, such as Hu et al., 2021, and Li et al., 2021, about the de-oxygenation problems in this region, provide background and justification of hypoxia-related study in PRE. Nevertheless, I feel this manuscript fails to connect the new data with the findings of these existing studies and covers only a small portion of the PRE; thus, its potential to convey what can be learned from a regional study to a broader audience is limited.

   A critical flaw of this study is the representativeness of the stations that all statistical analyses are built upon. I acknowledge these are valuable (30 yr) monthly data covered by these stations, yet their spatial coverage is mainly surrounding the Hongkong island; it is a challenge to draw any solid conclusion regarding the PRE, even the east part, without a throughout cross-reference with the model and data by the previously mentioned two studies. Instead of using these stations to refer to east PRE, I would say the author should do the opposite—to provide a possible water quality study of the Hongkong coast with impacts from the PRE.

   Built on the above point, I see in line 103 (page 4), "the observed DO profiles were interpolated by MATLAB along with the three subregions with a grid resolution of 600 m (length) and 0.3 m (depth)." But take a look at the distribution of the stations; they are concentrated mainly in the nearshore area surrounding Hongkong; I doubt they can be representative of the condition in the east part of the PRE (e.g., the ECTZ defined by Li et al., 2021). And any conclusion based on such "hypoxia area" analysis (e.g., Fig. 4 ) is thus questionable.

Response: As the reviewer pointed out, there have been many observational and modelling studies on hypoxia in the PRE over recent years. However, most of these studies focused on short-term oxygen dynamics (e.g. Li et al., 2021; Wang et al., 2017; Yin et al., 2004; Cui et al., 2018; Shi et al., 2019), whereas the long-term oxygen pattern was less investigated. Hu et al. (2021) had an attempt to elucidate the long-term evolution of low-oxygen conditions in the PRE using observations during 1976-2017 collected from multiple datasets; however, as they mentioned, there existed data gaps in certain years and lack of conformity in observational coverage, and the observations compiled were under sampled in several years, which brought out great obstacles for quantifying the long-term oxygen changes due to the data limitations (Hu et al., 2021). Given this situation, we decided to utilize the monthly data collected in the coastal waters off Hong Kong, which have significant merits in terms of temporal coverage (over three decades) and consistency of sampling locations. These coastal data with good spatiotemporal continuity enabled us to better estimate the long-term and interannual variations in low-oxygen conditions without concerns on the uncertainties that would arise from the usage of different data sources. Besides, various oxygenrelated parameters (e.g., chlorophyll-*a* concentrations) were measured as well and could be used to discern the key factors controlling the long-term oxygen changes.

Moreover, based on in-situ observations in the PRE, low-oxygen conditions exhibited large spatial variations. Low-oxygen conditions were observed in the upstream zone of the PRE (Li et al., 2020), Modaomen Bay, and Huangmaohai Bay (Wang et al., 2017; Yin et al., 2004; Cui et al., 2018; Shi et al., 2019). Besides, in the coastal waters off Hong Kong, prominent low-oxygen conditions at the bottom together with surface phytoplankton blooms have been frequently reported (Li et al., 2021; Li et al., 2020; Qian et al., 2018; Lu et al., 2018). Previous studies revealed that the dominant deoxygenation mechanism varied among subregions in the PRE. While the upper PRE (Li et al., 2020) and Modaomen Bay (Wang et al., 2017; Wang et al., 2018) were more heavily influenced by terrestrial organic matter, low-oxygen events in the Hong Kong waters were largely regulated by the joint effect of physical and eutrophic processes (Li et al., 2021; Qian et al., 2018). Therefore, the quantitative investigation on deoxygenation in the coastal waters off Hong Kong, which was close to the hotspot area of low-oxygen conditions in the eastern PRE (Figure r1), could be a significant supplement to the understanding of the low-oxygen conditions for the whole PRE. We agreed that Hong Kong data was indeed inappropriate to represent the whole situation of the eastern PRE, which needs more data covering boarder range (note that such data with long-term time series are not available at present). Thus, we have revised the phase "the eastern PRE" to "the coastal waters off Hong Kong" in the revised manuscript for more precise description. Nevertheless, we would like to emphasize the significance of our study, in which we tried to provide a better insight into the long-term oxygen changes and the underlying mechanisms for an important subregion in the PRE by utilizing this valuable dataset, especially from a quantitative perspective. Our results could contribute to the low-oxygen management in the lower reach of the PRE.

Based on the reviewer's comment, we have revised the manuscript as follows.

(1) In Page 3, Line 76-90 of the revised manuscript, we have added more descriptions of water quality and oxygen background information in different subregions, and have enhanced the descriptions regarding the current status and existing problems in the long-term low-oxygen research in the PRE to emphasize the significance of our work:

*"In this study, we utilize observational oxygen and related data collected by the Hong Kong Environmental Protection Department (HKEPD) at certain coastal sites off Hong Kong (see details in section 2.1 below), which have significant merits in terms of temporal coverage (~30 years) and consistency of sampling locations, to perform a quantitative analysis on the long-term oxygen changes (trend and interannual variability) in the region. Moreover, we also aim to discern the key factors controlling the interannual variability and long-term trends in the low-oxygen conditions and to quantify the relative contribution of each primary factor using multiple regression models (Murphy et al., 2011; Forrest et al., 2011; Wang et al., 2021). It is important to note that the HKEPD data with good spatiotemporal continuity allowed us to better estimate the long-term deoxygenation*

*in the coastal waters off Hong Kong, which was close to a hotspot area of low-oxygen conditions in the eastern PRE (Hu et al., 2021) and subject to frequent occurrences of low-oxygen and hypoxic events as well (Yin et al., 2004; Su et al., 2017; Shi et al., 2019). In addition, previous studies have showed that the dominant deoxygenation mechanisms varied between subregions in the PRE; for instance, the low-oxygen conditions in Modaomen Bay were primarily determined by terrestrial pollutant inputs (Li et al., 2020; Wang et al., 2017; Wang et al., 2018), whereas those in the coastal waters off Hong Kong were largely controlled by the joint effect of physical processes (e.g. convergence of water masses (Li et al., 2021)) and eutrophication (Qian et al., 2018). Therefore, the extensive investigation on deoxygenation performed here for the Hong Kong waters is a significant supplement to the understanding of low-oxygen conditions for the whole PRE."*

(2) In Page9-10, Line 266-308 of revised manuscript, we have added more discussion on the effects of wind, freshwater discharge, and nutrient loads on deoxygenation in combination with the relevant findings of the previous studies, and have enhanced the analysis on differences of key deoxygenation mechanism in different subregions of the PRE to improve the implication of our work:

*"It was noted that the monthly average wind direction in summer were generally southerly with small changes (mostly varying between 150° and 200°, Figure A1). Overall, our results indicated that the wind speed played a more important role in regulating the low-oxygen conditions in the coastal waters off Hong Kong from an interannual perspective, although the wind direction could significantly influence the short-term generation and development of low-oxygen conditions by modulating the Pearl River plume and material fluxes (Yin et al., 2004; Li et al., 2021).……*

*It has been widely recognized that eutrophication stimulated by anthropogenic nutrient inputs could provide a large quantity of depositing detritus and subsequently led to substantial oxygen depletion and occurrence of low-oxygen events (Rabalais et al., 2010; Fennel and Testa, 2018); for example, in the northern Gulf of Mexico (Feng et al., 2012; Forrest et al., 2011) and Chesapeake Bay (Wang et al., 2015), the interannual hypoxic areas in summer were directly regulated by the nutrient levels. Similar situation was found in the PRE (Li et al., 2020) and Hong Kong waters, as confirmed by the significant positive correlation between DIN and LOI (r = 0.65, p < 0.01)……"* (please see details in the revised manuscript)

Overall, we hope to emphasize our aim in exploring the long-term low-oxygen change and its important drivers for a significant part of the PRE, in which our analysis methods exhibited general applicability for further studies.

[Figure]

Figure r1. Observation sites during 1976-2014 in the PRE in previous studies (Hu et al., 2021) and station sites in our work (a). Note that black dots represent observation sites in previous studies and blue, red, green dots represent station sites in the northwestern, southern, and eastern subregions of Hong Kong waters, respectively. The incidence of low-oxygen (b1) and oxygen-deficiency (b2) conditions at the bottom of the PRE in summer during 1976-2014 (Hu et al., 2021).

2. In addition, the water depth of these stations varies from 5 m to more than 30m, and the analysis in this manuscript concentrated mostly surface and bottom, which impair the reliability of such analysis in the de-oxygenation study, which is very sensitive to water depth and vertical distribution of variables.

Response: We agreed that the analysis of deoxygenation and its related processes was sensitive to water depth and vertical distribution of variables. In fact, we did concern with this issue, especially for DO. That is exactly the reason why we used several vertical-profile-based metrics (e.g. the cross-sectional area and the layer thickness of low-oxygen conditions) to depict the long-term oxygen changes in the region (please see detailed descriptions in the Methods section 2.2). For example, as for the spatial extent of low-oxygen conditions, we estimated the vertical low-oxygen area and thickness by interpolation using the data at the surface, middle, and bottom layers according to the depth of each station. In addition, we have also investigated the occurrence of low-oxygen and hypoxic events at the surface, middle, and bottom layers (please see Figure A3 in the revised manuscript), which indicated that low-oxygen events mainly appeared in the bottom waters of summer. Vertical distributions of DO in summer was provided as well (please see Appendix Figure A4 in the revised manuscript). For other variables, we also performed a thorough analysis on their spatial (surface, middle, bottom layers) and temporal patterns. We found that the key problems were concerned mostly in the surface and bottom waters. For instance, while low DO mostly occurred at the bottom, high Chl $a$ concentrations frequently occurred at the surface. Therefore, in order to embody the compactness of analysis and data display, we only exhibited the long-term variations of water quality constituents and DO concentrations at the surface and bottom layers (e.g. results shown in Figures 2-3). Nevertheless, we would like to emphasize that several oxygen-related

metrics incorporating information of vertical DO distributions were analyzed and discussed throughout the manuscript, including the Low-Oxygen Index (LOI as defined in section 2.2) and the cross-sectional area of low-oxygen conditions.

3. In the abstract, the author indicated that "there is still a lack of quantitative understanding of the long-term trends and interannual variabilities in oxygen conditions in the PRE as well as the driving factors, which was comprehensively investigated in this study," which I could not agree, I think Hu et al. and Li et al. provided good studies about the mechanism of oxygen dynamics in the PRE. Yet, this manuscript fails to connect what is observed by the stations surrounding Hongkong to what has been reported in a larger geospatial content (PRE and coastal/shelf water).

Response: For the sentence *"there is still a lack of quantitative understanding of the long-term trends and interannual variabilities in oxygen conditions in the PRE as well as the driving factors, which was comprehensively investigated in this study"* that we mentioned in manuscript, we would like to provide more explanations. As we mentioned earlier (please see our response to Comment #1), there were indeed many studies about the mechanism of hypoxic generation and development in the PRE, but most of them focused on short-term oxygen dynamics (e.g. Li et al., 2021; Wang et al., 2017; Yin et al., 2004; Cui et al., 2018; Shi et al., 2019), while the long-term mechanism received less attention. Hu et al. (2021) had an attempt to elucidate the long-term evolution of low-oxygen conditions. However, as mentioned in their analysis, more quantitative perspective on hypoxic mechanism was needed, limited by the lack of accessible continuous observations. Therefore, the long-term analysis of Hong Kong data could make up for these problems to some extent as mentioned earlier (please see our response to Comment #1).

With respect to the connection of our analysis with previous studies, we also believed that it was vitally necessary. Although most of researches of low-oxygen focused on the whole PRE, we tried to compare their results involved in the Hong Kong waters with our findings. For example, as for the effect of wind (in section 4.1), we explained our results by combining the findings of previous studies (as already mentioned in Line 240-245 of the original manuscript). Nevertheless, we agreed that this part of discussion should be enhanced. As we mentioned earlier (please see our response to Comment #1), we have provided more detailed discussions as follows:

(1) In Page 9, Line 271-282 of the revised manuscript, we have enhanced the comparison on the effect of wind between our results and previous studies:

*"However, the weak correlation between WS and $\Delta\rho$ (Figure 8) indicated that the wind forcing may control hypoxia through other alternative mechanisms. Actually, weak winds in combination with flow convergence induced by wind-driven circulation could contribute to long water residence time and nutrient accumulation in the eastern PRE and thus favored the phytoplankton blooms (Li et al., 2021). This could be supported by the significant negative correlation between WS and Chl a*

*(r = -0.62, p < 0.01). In contrast, the wind direction showed less significant effect on the interannual variability in low-oxygen conditions, as suggested by the comparatively poor performance in the LOI fitting and the weaker correlations of the wind direction-related variables with LOI (Table A3). It was noted that the monthly average wind direction in summer were generally southerly with small changes (mostly varying between 150° and 200°, Figure A1)……"*

(2) In Page 9, Line 282-289 of the revised manuscript, we have added the reference the effect of nutrients on interannual variation in low-oxygen conditions in previous studies:

*"It has been widely recognized that eutrophication stimulated by anthropogenic nutrient inputs could provide a large quantity of depositing detritus and subsequently led to substantial oxygen depletion and occurrence of low-oxygen events (Rabalais et al., 2010; Fennel and Testa, 2018); for example, in the northern Gulf of Mexico (Feng et al., 2012; Forrest et al., 2011) and Chesapeake Bay (Wang et al., 2015), the interannual hypoxic areas in summer were directly regulated by the nutrient levels. Similar situation was found in the PRE (Li et al., 2020) and Hong Kong waters, as confirmed by the significant positive correlation between DIN and LOI (r = 0.65, p < 0.01)."*

(3) In Page 10, Line 295-301 of the revised manuscript, we have enhanced the comparison of the effect of freshwater discharge on the interannual variation in low-oxygen conditions between our results and previous studies:

*"Due to its long distance from the river outlets of the Pearl River, the coastal waters off Hong Kong were relatively less influenced by terrestrial inputs (Yu et al., 2020) and the effect of freshwater discharge and its carrying organic matter in this area was not as significant as that in other subregions (e.g. the upper reach of Lingdingyang Bay and the western PRE). Nevertheless, freshwater discharge in combination with the wind-driven circulation could significantly affect the water residence time (Sun et al., 2014) and nutrients accumulation in the Hong Kong waters (Li et al., 2021). Specifically, the weakened discharge could prolong the retention of nutrients and thereby stimulate local productions of organic matter in the region (Li et al., 2021), which ultimately promoted oxygen depletion."*

4. The author uses wind speed in their statistic analysis which is also questionable. How about wind direction, and does the wind play the same role in low oxygen development over a year? I am asking because Li et al. (2021) indicated that both wind direction and intensity influenced the circulation nutrient flux, detritus, and vertical mixing. Also, as suggested by Li et al., 2021, what is the role of shelf circulation in physics (mixing, etc.) and nutrient and sediment delivery? Li et al. (2021) and Feng et al. (2014) show that the upwelling and downwelling favorable wind condition has different impacts on the low-oxygen development in this area. It is problematic to use the monthly mean wind speed as a predictor without looking into wind's detailed role in this environment.

Response: In fact, during the preparation of our study, we have also tried to examine the effect of

wind direction since the transport of nutrient and detritus, vertical mixing, and residence time could be regulated by wind-driven circulation (Li et al., 2021). Previous studies have also revealed that the southwestern (upwelling-favorable) wind could enhance hypoxia in the eastern PRE (Li et al., 2021). Therefore, we processed the daily data of wind speed and direction into monthly average wind speed (WS), southwestern wind duration (SWWD), southwestern wind cumulative stress (SWCS), and southeastern wind cumulative stress (SECS) in summer (please see Figure r9 below), and then we carried out a suite of multiple regressions to fit the Low-oxygen Index (LOI) for each wind-related variable. As shown in Table r4 below, the fitting effect of LOI was better using WS, which also has the highest correlation with LOI among the wind-related variables. We considered that WS was the most significant explanatory wind-related factor for the interannual variations in LOI, and thus we eventually selected WS to be the wind-related input variable in the multiple regression. Besides, we noted that the long-term trends of direction-related variables (i.e. SWWD, SWCS, and SECS) were not as significant as WS (please see Figure r9 below). This probably suggested that the wind direction-related processes played an important role in short-term low-oxygen events, but contributed less from the interannual and long-term perspective.

Based on the reviewer's concern on the wind direction, we have provided more discussion on the selection about wind variables in Page 5, Line 138-146 of the revised manuscript as follows:

*"As for the selection of the wind variable in use, the daily wind data were processed into monthly average wind speed (WS), southwestern wind duration (SWWD), southwestern wind cumulative stress (SWCS), and southeastern wind cumulative stress (SECS) in summer (June-August) to examine the effect of wind speed and direction. Then, a suite of multiple regressions was carried out to fit the LOI for each wind-related variable. As shown in Table A3, the fitting effect of LOI was better when using WS, which also has the highest correlation with LOI among the wind-related variables, revealing that WS explained the most interannual variation of LOI among the wind-related factors. Therefore, WS was eventually adopted to be the wind-related input variable in the multiple regression with freshwater discharge (flow), the monthly spatial-average of bottom temperature (T), and surface DIN in the summer."*

Besides, we have enhanced the discussion about the effect of wind speed and direction in Page 9, Line 275-282 of the revised manuscript as follows:

*"In contrast, the wind direction showed less significant effect on the interannual variability in low-oxygen conditions, as suggested by the comparatively poor performance in the LOI fitting and the weaker correlations of the wind direction-related variables with LOI (Table A3). It was noted that the monthly average wind direction in summer were generally southerly with small changes (mostly varying between 150° and 200°, Figure A1). Overall, our results indicated that the wind speed played a more important role in regulating the low-oxygen conditions in the coastal waters off Hong Kong from an interannual perspective, although the wind direction could significantly influence the short-term generation and development of low-oxygen conditions by modulating the*

*Pearl River plume and material fluxes (Yin et al., 2004; Li et al., 2021)."*

Please note that we have provided Figure r9 and Table r4 (please see below) in the Appendix of the revised manuscript:

[Figure]

Figure r9. Average wind speed (WS), southwestern wind duration (SWWD), southwestern wind cumulative stress (SWCS), southeastern wind cumulative stress (SECS), average wind direction (WD) in summer, and their long-term trends during 1994-2018. Note that the negative values of SWCS and SECS represent southwestern and southeastern wind, respectively. The trends and significant p values were shown in the title of each subgraph.

Table r4. Coefficients of determination ($R^2$) for average wind speed (WS), southwestern wind duration (SWWD), southwestern wind cumulative stress (SWCS), southeastern wind cumulative stress (SECS) in fitting LOI; Pearson correlation coefficient of WS, SWWD, SWCS, and SECS with LOI. Note that the symbols * and ** represents the significant level at $p < 0.05$ and $p < 0.01$, respectively.

| Fitting cases | $R^2$ of fitting LOI | Variables | Correlation with LOI |
|---|---|---|---|
| WS+flow+T+DIN | 0.61 | WS | -0.67** |
| SWWD+flow+T+DIN | 0.55 | SWWD | 0.48* |
| SWCS+flow+T+DIN | 0.55 | SWCS | -0.33 |
| SECS+flow+T+DIN | 0.57 | SECS | 0.25 |

5.  The author's conclusion that the eastern PRE would "develop into a severe hypoxic state within the next two decades" is too strong to be supported by the analysis provided by this study. For instance, what do the wind, large-scale circulation (cause it affects lateral delivery of water and nutrient, etc.), and river (Pearl plus wastewater from the city) look like in the next two decades? The Pearl River discharged into the PRE from the north, yet if we focus on the spatial scale covered by the stations in this study, what is Pearl River's role in MM stations? Also, what is the impact of overland runoff from Hongkong, such as wastewater discharge, which is also indicated by Hu et al. (2021), and the author briefly mentioned this in Line 250 of page 8.

Response: We would like to make more explanation for the prediction of low-oxygen conditions. Based on the observational data and the multiple regression model, we have discerned the contributions of wind speed, freshwater discharge, water temperature, and DIN to the long-term deoxygenation, in which wind speed and DIN contributed the most. Our results showed that the wind speed was expected to decrease at a rate of 0.6 m/s within two decades, which would contribute to longer water residence time and nutrient accumulation (Li et al., 2021) in combination with wind-driven circulation, favoring the generation and development of low-oxygen conditions. Likewise, DIN was expected to increase by 0.2 mg/L in two decades according to the observed growth trend, which would enhance eutrophication and subsequent low-oxygen condition within the region. Overall, based on the long-term trends of each influential variable and the proposed regression model, we predicted that low-oxygen conditions would develop into a severe hypoxic state. As we mentioned in Page 10, Line 312-314 of the revised manuscript, based on the observed deoxygenation rate, the bottom DO minimum was expected to decrease by approximately 15%-70% in 5-20 years (reaching a level of 0.4-1.6 mg/L) compared to the climatological mean of 1994-2018. Certainly, this inference was laid on the assumption that wind speed, freshwater discharge, water temperature, and DIN would keep increasing/decreasing at the same rate in the next two decades, and the reality situation would develop more complicatedly, for example, nutrient management strategies would be implemented and climate factors would change in a non-linear pattern. However, our findings still served as an alarming signal that changes in wind and freshwater discharges could cancel out potential benefits of nutrient management. To this end, it is of great importance to conduct long-term and more intensive control on nutrient inputs in order to mitigate the low-oxygen conditions in the region.

Moreover, our results showed that the long-term changes in low-oxygen conditions were ultimately driven by the long-term changes in physical factors (e.g. wind and circulation) and biogeochemical factors (e.g. nutrients), and the proposed regression model (fitting the LOI) actually implicitly incorporated the influence of physical-biogeochemical processes on DO to some extent. We have also tried to discuss the effects of wind, freshwater discharge, and DIN in combination with the circulation and nutrient fluxes in section 4.1 and 4.2, but we realized that it was hard to

explore the detailed changes in mechanic processes clearly based on observational data and statistic methods only. To this end, further studies such as numerical simulation are needed in the future. Based on the reviewer's comment, we have improved this part of discussion by adding the rationale of our prediction and the implication of our work in Page 11, Line 340-352 of the revised manuscript as follows:

*"Similarly, the long-term oxygen changes in terms of the areal extents and arrival timing of hypoxia have also been found in other coastal systems. For example, in Chesapeake Bay, sea level rise and elevated freshwater discharges would lead to an approximately 10%-30% increase in hypoxic volume between the late 20th and the mid-21st centuries (Ni et al., 2019), while the increase in water temperature would cause hypoxia to develop 5-10 days earlier in ~30 years (Ni et al., 2020). In the northern Gulf of Mexico, the growth in riverine nutrient inputs would result in an increase in the frequency of hypoxia occurrence by 37% (Justić et al., 2003). While in the Hong Kong waters, low-oxygen conditions would develop into hypoxic conditions in two decades with larger areal extent and earlier arrival ascribed to the ongoing alterations in physical conditions and nutrients as mentioned earlier. This inference was based on the assumption that the external factors (e.g. wind speed, DIN, discharges) would change at the same rates as those in the past 25 years. Although the real situation would be more complicated and compounded by factors such as the implementation of management and non-linear changes in climatic factors, our findings still served as an alarming signal that changes in wind and freshwater discharges could cancel out potential benefits of nutrient management. To this end, it is of great importance to conduct long-term and more intensive control on nutrient inputs in order to mitigate the low-oxygen conditions in the region."*

For the effect of wastewater, changes in DIN levels over the past 25 years were influenced by physical conditions and anthropogenic inputs (e.g. from the Pearl River and Hong Kong local pollution). Actually, the changes of local wastewater could be reflected in the variations of DIN. Therefore, we didn't distinguish the influence from the Pearl River and local pollution in our study when we aimed to investigate the effect of the long-term variations of nutrients levels from a general perspective. As for the role of the Pearl River, there were indeed differences among stations, which were also shown by different correlations of DO with other external variables in each station. It would be interesting to dig into this issue, but our present work mainly focused on the whole status of low-oxygen conditions in the selected region, rather than dealing with the spatial differences.

**Reference**

Cui, Y., Wu, J. X., Ren, J., and Xu, J.: Physical dynamics structures and oxygen budget of summer hypoxia in the Pearl River Estuary: Dynamical structures and oxygen budget of hypoxia, Limnology and Oceanography, 64, 10.1002/lno.11025, 2018.

Hu, J., Zhang, Z., Wang, B., and Huang, J.: Long-term spatiotemporal variations and expansion of low-oxygen conditions in the Pearl River estuary: A study synthesizing observations during 1976–2017, 10.5194/bg-2020-480, 2021.

Li, D., Gan, J., Hui, C., Yu, L., Liu, Z., Lu, Z., Kao, S.-j., and Dai, M.: Spatiotemporal Development and Dissipation of Hypoxia Induced by Variable Wind-Driven Shelf Circulation off the Pearl River Estuary: Observational and Modeling Studies, Journal of Geophysical Research: Oceans, 126, e2020JC016700, https://doi.org/10.1029/2020JC016700, 2021.

Li, X., Lu, C., Zhang, Y., Zhao, H., Wang, J., Liu, H., and Yin, K.: Low dissolved oxygen in the Pearl River estuary in summer: Long-term spatio-temporal patterns, trends, and regulating factors, Marine Pollution Bulletin, 151, 110814, https://doi.org/10.1016/j.marpolbul.2019.110814, 2020.

Lu, Z., Gan, J., Dai, M., Liu, H., and Zhao, X.: Joint effects of extrinsic biophysical fluxes and intrinsic hydrodynamics on the formation of hypoxia west off the Pearl River Estuary, Journal of Geophysical Research: Oceans, 123, 6241-6259, 2018.

Qian, W., Gan, J., Liu, J., He, B., Lu, Z., Guo, X., Wang, D., Guo, L., Huang, T., and Dai, M.: Current status of emerging hypoxia in a eutrophic estuary: The lower reach of the Pearl River Estuary, China, Estuarine, Coastal and Shelf Science, 205, 58-67, https://doi.org/10.1016/j.ecss.2018.03.004, 2018.

Shi, Z., Liu, K., Zhang, S., Xu, H., and Liu, H.: Spatial distributions of mesozooplankton biomass, community composition and grazing impact in association with hypoxia in the Pearl River Estuary, Estuarine, Coastal and Shelf Science, 225, 106237, https://doi.org/10.1016/j.ecss.2019.05.019, 2019.

Wang, B., Hu, J., Li, S., and Liu, D.: A numerical analysis of biogeochemical controls with physical modulation on hypoxia during summer in the Pearl River estuary, Biogeosciences, 14, 2979-2999, 10.5194/bg-14-2979-2017, 2017.

Wang, B., Hu, J., Li, S., Yu, L., and Huang, J.: Impacts of anthropogenic inputs on hypoxia and oxygen dynamics in the Pearl River estuary, Biogeosciences, 15, 6105-6125, 10.5194/bg-15-6105-2018, 2018.

Yin, K., Lin, Z., and Ke, Z.: Temporal and spatial distribution of dissolved oxygen in the Pearl River Estuary and adjacent coastal waters, Continental Shelf Research, 24, 1935-1948, https://doi.org/10.1016/j.csr.2004.06.017, 2004.

---

## Author Response (AR2)

**Editorial comments:**

Your revised manuscript was reviewed by one of the original reviewers. The reviewer was generally positive (minor revisions). I concur with the reviewer's comments. Please address the comments offered by the reviewer on the revised (second version) of the manuscript and return as a third version (second revision).

Response: For the second revision of our manuscript, we would like to thank the editor and the referee for the positive comments and suggestions. We have revised the manuscript as suggested. Detailed responses to all comments are given below (responses are shown in blue and relevant changes are marked in red in the revised manuscript).

**Anonymous Referee #1**

**Comments**

1. The authors made helpful changes to this manuscript, including describing how the sampling locations fit within the broader region, adding more details to the statistical methods descriptions, and adding more comparison to other studies. I have a few suggestions to finalize the changes. Be sure to check all figure references in the text because I found a few that were incorrect due to the addition of new figures.

Response: Thank you for the positive comment. As suggested, we have checked all figure references in the revised manuscript. Please see details in the point-to-point response below. In addition, we have also double checked the figures in Appendix and reordered Figures A4, A1, A5, A2, and A3 to Figures A1, A2, A3, A4, and A5 in the revised manuscript (second revision). The corresponding figure references have also been revised in the text.

2. Your changes in naming the location seem appropriate to me. However, check the grammar in your edits:
   Line 15, abstract: Change "the subregion of the PRE" to "a subregion of the PRE"
   Line 27, abstract: Change the last word from "has" to "have"

Response: We have revised these sentences in Page 1, Line 15 and Line 28 of the revised manuscript as follows:

   *"Therefore, the long-term deoxygenation in a subregion of the PRE (the coastal waters off Hong Kong) was comprehensively investigated in this study using monthly observations during 1994-2018."*

   *"In summary, the Hong Kong waters have undergone considerable deterioration of low-oxygen conditions driven by substantial changes in anthropogenic eutrophication and external physical factors."*

3. Intro, Line 54 – change "ocean" to "estuary" here again to represent the citation correctly. Or just delete the word.

Response: As suggested, we have deleted the word "ocean".

4. Line 120. For the sentence that starts with "Moreover…", I'm still trying to understand if each's months data was interpolated separately. It sounds like it was, but this sentence could be clearer like: "Moreover, the same treatment procedure was applied to the data in each month for 25 years to generate an interpolation set for every month."

Response: Data in each month was interpolated separately. As suggested, we have revised the sentence in Page 4, Line 120 of the revised manuscript as follows:

*"Moreover, the same treatment procedure was applied to the data in each month of 25 years to generate an interpolation set for every month, making it consistent when investigating the interannual variations in low-oxygen conditions."*

5. Line 202. I think this reference should be to Figure A3. And I'd suggest removing the 10% statement because it is unclear what it refers to.

Response: Please note that we have reordered the figures in Appendix. We have corrected this reference mistake and revised the statement in Page 7, Line 201 of the revised manuscript as follows:

*"Statistic results showed that low-oxygen events mainly appeared in the bottom waters of summer, which had much higher occurrences of DO < 4 mg/L and DO < 2 mg/L compared to other seasons and other layers (Figure A5)."*

6. Line 215. This reference to Appendix Figure A2 seems to be the wrong figure reference.

Response: Indeed, it was a wrong figure reference. It should be referred to Figure A1. We have corrected this mistake in the revised manuscript.

7. Figure 4. An addition to the caption is still needed to define the grey region in Figure 4a, and/or put it in the legend at the top.

Response: The grey patch in Figure 4a represents the range of bottom DO observed at the 10 stations in three summer months. We have revised the caption in Page 20, Line 550 of the revised manuscript as follows:

*"Figure 4. Interannual variations in the spatiotemporally (10 stations in June, July and August) mean and minimum concentrations of observed DO at the bottom (a), the cross-sectional areas (b) and layer thicknesses (c) of low-oxygen conditions, and LOI (e) in summer during 1994-2018. Note that the grey patch in (a) represents the range of bottom DO observed at the 10 stations in three summer months; the coloured bars in (b) and (c) show the mean values of three summer months, while the black thin error bars represent the range across three summer months."*

8. Response to comment 13. I appreciate your description of the wind speed decreases and agree it is supported as a true condition in the literature. I suggest adding a sentence about this to the text. It could likely be just one sentence that notes that average wind speeds have decreased, and cite the sources that discussed this.

Response: We have added a brief description on wind speed decreases in the Pearl River Basin and the northern South China Sea and the citations in Page 10, Line 315 of the revised manuscript as follows:

*"It was noted that WS exhibited a decreasing trend of 0.03 m/s per year (p < 0.05, Figure 9a) within the coastal regions off Hong Kong over the past 25 years, while similar situation was also found in the Pearl River Basin (Zhang et al., 2019) and the northern South China Sea (Gao et al., 2020) due to the long-term climate changes (Xu et al., 2006; Zhang et al., 2009; Chen et al., 2020)."*

---

## Author Response (AR3)

Response: Thank you for accepting our manuscript. There was a mistake in figures submitted before. In Figure 1, we incorrectly typed the city name "Shenzhen" into "Shenzheng". We have corrected it and uploaded the latest figures ZIP this time.